# Compost Quality Indexes (CQIs) of Biosolids Using Physicochemical, Biological and Ecophysiological Indicators: C and N Mineralization Dynamics

**Héctor Iván Bedolla-Rivera** [1], **Eloy Conde-Barajas** [1,2], **Sandra Lizeth Galván-Díaz** [1], **Francisco Paúl Gámez-Vázquez** [3], **Dioselina Álvarez-Bernal** [4] **and María de la Luz Xochilt Negrete-Rodríguez** [1,2,*]

1   Departamento de Posgrado de Ingeniería Bioquímica, TNM/IT de Celaya, Ave. Tecnológico y A. García Cubas No. 600, Celaya 38010, Guanajuato, Mexico
2   Departamento de Ingeniería Bioquímica y Ambiental, TNM/IT de Celaya, Ave. Tecnológico y A. García Cubas No. 600, Celaya 38010, Guanajuato, Mexico
3   Campo Experimental Bajío, INIFAP, Carretera Celaya San Miguel de Allende Kilómetro 6.5, Celaya 38010, Guanajuato, Mexico
4   Instituto Politécnico Nacional, Centro Interdisciplinario de Investigación para el Desarrollo Integral Regional, Unidad Michoacán, Justo Sierra No. 28, Centro, Jiquilpan 59510, Michoacán, Mexico
*   Correspondence: xochilt.negrete@itcelaya.edu.mx; Tel.: +52-461-200-9580

**Abstract:** The increasing production of biosolids (BS) as a result of urban wastewater treatment generates pollution problems in their management and final disposal, and a better management is needed for their disposal. The composting of BS is an alternative process for obtaining a product with potential application as an organic amendment in the recovery of agricultural soils. As a biotechnological contribution, this study analyzed a composting process with BS, bovine manure (BM) and rice husks using four treatments T1 (C/N = 24); T2 (C/N = 34); T3 (C/N = 44); T4 (C/N = 54) for 120 days, in order to develop compost quality indexes (CQIs) through the analysis of 18 physicochemical, biological and ecophysiological indicators. Subsequently, three methodologies—successfully used on soils—were implemented for the development of the CQIs called "unified", "additive" and "nemoro". The indicators that comprised the CQIs were nitrification index (NI) and synthetic enzymatic index (SEI). The CQIs made it possible to differentiate the quality of the compost according to the treatments applied. The treatments used resulted in composts considered phytonutritious whose average quality value depending on the CQI developed was considered high ($CQI_w$ = 0.62), moderate ($CQI_a$ = 0.56) and low ($CQI_n$ = 0.30). The developed CQIs can be applied to determine the quality of BS composting systems reducing the cost of monitoring.

**Keywords:** biosolid; compost; enzyme activity; principal component analysis; unified index; additive index; nemoro index; C and N transformation

## 1. Introduction

Water pollution is an issue of environmental, social and economic concern. The growing population, urbanization and industrialization worldwide has led to a rise in the concentration of pollutants in water bodies [1]. For this reason, the creation of wastewater treatment plants (WWTPs) has increased in order to reduce the load of pollutants, aiming for the protection of the environment and the health of the inhabitants.

Mexico is no exception to this problem. Most WWTPs apply biological treatment for the sanitation process using activated sludge, obtaining treated water and a quantity of stabilized sludge called "biosolids" (BS) [2]. Water treatment capacity at a national level has been rising over the last decade, having 2766 WWTPs with an installed treatment capacity of 196,000 L s$^{-1}$ by the year 2020 [3]. Increased water treatment capacity consequently raises the production of BS, causing it to accumulate on land adjacent to the facilities or to be deposited in landfills, requiring an increasing amount of land for its final disposal [4]. Most

of the BS produced in Mexico is disposed of in landfills. However, BS is characterized by a composition rich in nutrients and organic matter (OM) [5]. These BS can be used as organic amendments for the remediation of degraded soils or as organic fertilizers for agricultural soils, improving their physical, chemical and biological properties [6]. Improved soil conditions have been reported following the addition of BS, through improved water holding capacity (WHC) and improved soil structure, as well as increased fertility from the addition of easily degradable C and N compounds [7]. The addition of BS to degraded soils increases plant cover, microbial activity and diversity, which are stimulated by the addition of easily degradable OM, allowing better nutrient cycling in the soil [5,8–10].

The application of BS in soils in Mexico is controversial, due to the social stigma of the process from which they are obtained. In order to ensure the safety of BS, national and international regulations are in force (NOM-004-SEMARNAT-2002 and EPA 503) [11,12], which establish limits on the concentration of contaminants and biological agents—bacteria, fungi and actinomycetes—when applying BS to soils.

In addition to the aforementioned regulations, another process to ensure the safety of BS is composting [13], a process by which most pathogens are eliminated through decomposition processes at temperatures of around 55 °C. The correct composting process will result in a stable, mature and nutrient-rich compost to use as an organic amendment in degraded soils and crops. However, the intrinsic variability of the process of obtaining BS leads to variability in the composting process [13]. For this reason, constant monitoring by means of different physicochemical and biological indicators is necessary in order to ensure a quality end product.

As composting is a complex and multifactorial process, monitoring can be laborious and costly. Therefore, several studies have established a small number of indicators related to the maturation, stability and quality of compost products. Saldarriaga et al. [14] suggest that four indicators—respirometry, ash content, moisture content and WHC—are sufficient to determine the quality of composts obtained from municipal solid waste. Peña et al. [15] established quality indicators from a PCA—such as germination index (GI) and relative emergence (RE)—related to the phytotoxicity of composts in systems consisting of fruit residues, biosolids and frying residues. Meena et al. [16] defined five indicators highly related to the quality of composts made from various organic wastes, pressed sewage sludge and gypsum, added with inorganic sources of sulfur (S). The resulting indicators were water soluble sulfur (WSS), total nitrogen (TN), GI, NI (ammonium/nitrate ratio (N-NH$_4^+$/N-NO$_3^-$)), hydrogen potential (pH) and the concentration of arylsulfatase. Those indicators were integrated into an equation weighed by the variability of the principal components (PCs) obtained from a PCA. In the same context, studies such as those described above have used a PCA to reduce the dimensionality of the composting process. However, they use BS as co-substrates in lower proportion in the composting processes than those established in the treatments of the present study, therefore there are no quality indicators directly related to BS composting. The aim of this study was to carry out an analysis of composting systems using BS as the main substrate—higher proportion substrate—for the establishment of physicochemical, biological and ecophysiological indicators related to compost quality, by means of a PCA and its subsequent integration in CQIs using three different techniques focused on the establishment of soil quality indexes. The hypothesis put forward for this study is that the implementation of techniques for establishing soil quality indexes in composts will allow for the selection of indicators related to the quality of BS compost, as well as to differentiate the treatments implemented based on the quality of the compost obtained.

## 2. Material and Methods

### 2.1. Obtaining and Conditioning of BS, BM and Rice Husk Samples

The BS was obtained from a WWTP with activated sludge treatment under extended aeration, located in the municipality of Celaya, Guanajuato, Mexico, with coordinates 20°29′34″ N, 100°56′03″ W. The BS sampling was systematic, taking samples every ten

minutes for a period of one hour, with handling and transport under sterile conditions at 4 °C. For physicochemical analyses, the BS samples were transported at room temperature (*T*). Subsequently, the samples were convective dried and sieved to a particle size of 2 mm. BM as co-substrate was obtained from "La Maceta" ranch located at coordinates 20°32′27″ N, 100°41′49″ W. The BM was obtained from Holstein and Indian cattle, fed on alfalfa and pasture. The BM samples were transported to the laboratory in plastic bags with subsequent drying at room *T*. For better particle size control, the samples were crushed to 5 mm. Dried rice husks were used as a bulking agent (BA) due to their high silica content, giving high mechanical resistance to microbial degradation [17].

## 2.2. Physicochemical Characterization of BS and BM

Heavy metals—As, Cd, Cr, Cu, Hg, Ni, Pb and Zn—from the BS and BM samples were characterized in triplicate (1.0 kg BS or BM) using the EPA method 3050B [12], reported in mg k$^{-1}$ of dry BS or BM. The pH was determined in a BS:water or BM:water solution (1:5 w v$^{-1}$) as stated by Thomas [18]. Electrical conductivity (EC) was determined in a BS:deionized or BM:deionized water solution (1:2.5 w v$^{-1}$) using a HANNA HI9811-5 digital conductivity meter(Woonsocket, RI, USA), reported in dS m$^{-1}$ [19]. Moisture (M) was determined by placing 2.0 g of BS or BM under forced convection (100 °C) in a RIOSSA H-33 (Monterrey, NL, Mexico)oven until having a constant weight, reported as a percentage [20]. For the determination of *T*, monitoring was carried out at the beginning and during the mineralization dynamics using the methodology of Tiquia et al. [21], reporting the results in °C. For bulk density (BD), BS or BM samples were placed in a 1.0 L test tube, compacted to reduce voids, then dried by forced convection at 105 °C and weighed. The weight of the BS or BM was divided by the volume of the specimen [22] and reported in g cm$^{-3}$. NT was determined by the micro-Kjeldahl method using a micro-Kjeldahl equipment model mdk-6, San Pedro Tlaquepaque, Jal, Mexico [23], quantified colorimetrically at 660 nm in a JENWAY 6305, (Sheung Wan, Hong Kong, China) single beam UV-Vis spectrophotometer, reported in mg N kg$^{-1}$ of dry BS or BM, using the same equipment for all spectrophotometric analyses. Inorganic N—expressed as ammonium (N-NH$_4^+$), nitrite (N-NO$_2^-$) and nitrate (N-NO$_3^-$)—was analyzed by pre-extraction of the samples with potassium sulfate (0.5 M) (1:5 w v$^{-1}$) for two hours [24]. After extraction, the extract was filtered using Whatman No. 2 filter paper (Solna, Sweden) and the filtrate was kept frozen at −4 °C until analysis. For N-NH$_4^+$, a salicylic acid solution (5% w v$^{-1}$) was used, and colorimetric determinations were performed at a wavelength of 660 nm, reported in mg N-NH$_4^+$ kg$^{-1}$ of dry BS or BM [25]. For the determination of N-NO$_2^-$, a diazonium salt solution (0.3% w v$^{-1}$) was used, quantified at a wavelength of 410 nm and reported in mg N-NO$_2^-$ kg$^{-1}$ of BS or dry BM [25]. For N-NO$_3^-$ analysis, a sulfanilamide solution (0.5% w v$^{-1}$) was used, quantified at a wavelength of 540 nm and reported in mg N-NO$_3^-$ kg$^{-1}$ of dry BS or BM [25]. Total organic C (TOC) was quantified colorimetrically at a wavelength of 660 nm and reported as a percentage [26]. Ammonia (N-NH$_3^+$) was determined by monitoring N volatilization using the method by Conde et al. [24]. For this purpose, 20.0 g of BS or BM samples were placed in 1.0 L glass bottles, incubated with 20.0 mL of boric acid (2%) contained in amber bottles at 25 °C. After three days, an aliquot of 5.0 mL of boric acid was taken by gauging to 50.0 mL with distilled water. Subsequently, the samples were titrated with sulfuric acid (0.02 N) and a mixed indicator, expressing the results in mg N-NH$_3^+$ kg$^{-1}$ of dry BS or BM. Soluble organic C (SOC) was estimated using a sample of 10.0 g of BS or BM in 25.0 mL of distilled water. Afterward, the samples were centrifuged at 2500 rpm for five minutes in a ZEIGEN CH90-1A, Ningbo, China centrifuge, and then gravity filtered. For the quantification of SOC concentration, 2.0 mL of filtrate was taken and analyzed spectrophotometrically at a wavelength of 590 nm, reported as mg SOC kg$^{-1}$ of dry BS or BM [27].

### 2.3. Experimental Design

The BS and BM composting units were designed based on the initial values of the TOC and TN (C/N) indicators (Table 2). Due to the nature of its high silica content, the BA was not included in the calculation of the C/N ratio. However, 30.0 g of BA was added to all composting units and treatments. The volumetric capacity of the composting units was 0.113 $m^3$ when adjusting the WHC to 40%. A completely randomized block experimental design (CRB) with four treatments and five replicates was applied. During the composting process, each composting system was sampled in triplicate on days 0, 7, 15, 30, 60, 90 and 120. *T* and M loss due to natural evapotranspiration were controlled during the whole composting process. In order to increase *T* in the thermophilic stage, the composting systems were covered with tulle 15 nylon fabric during the first two weeks of the experiment. M was maintained throughout the experiment at 40–60% of WHC by periodic manual application of sterile distilled water. Aeration was carried out by turning and manual-mechanical mixing every three days for the first two weeks, thereafter, the aeration process was carried out in periods of 8–15 days until the end of the experiments. The treatments (T) used in the composting systems are listed as follows: T1 (BS [C/N = 24]); T2 (BS and BM [C/N = 34]); T3 (BS and BM [C/N = 44]); T4 (BM [C/N = 54]).

### 2.4. Dynamics of C and N Mineralization in Composting Systems

#### 2.4.1. Physicochemical Indicators

The pH, EC, WHC, $N-NH_4^+$, $N-NO_2^-$, $N-NO_3^-$, $N-NH_3^+$, $N_{min}$, NI, $C-CO_2$ and SOC were analyzed in triplicate during the C and N mineralization dynamics. The methodologies used for the indicators pH, EC, M, *T*, $N-NH_4^+$, $N-NO_2^-$, $N-NO_3^-$, $N-NH_3^+$ and SOC were previously described in Section 2.2. The WHC was determined using the methodology described by Nannipieri [25]. Briefly, 20 g of dry compost was placed on Whatman No. 2 filter paper (Solna, Sweden), 100 mL of distilled water was added and left to stand for 24 h. The WHC was calculated by the difference between the weight of the filter with sample and the filter without sample, reporting the results as a percentage. The $N_{min}$ indicator—which represents the net N mineralization—was determined by the contribution of the inorganic N indicators ($N-NH_4^+ + N-NO_2^- + N-NO_3^-$) [28], reported in mg N $kg^{-1}$ of dry compost. The NI indicator—which would represent the partial flux of the inorganic N cycle in the composting system—was obtained from the quotient of the indicators $N-NH_4^+$ and $N-NO_3^-$, where high values have been related to denitrification and low values to nitrification processes [16]. Regarding C mineralization, this was monitored by the emission and evolution of carbon dioxide C from microbial activity. Finally, 20.0 g of a compost sample was placed in 1.0 L glass jars and incubated for three days together with a bottle containing 25.0 mL of sodium hydroxide (1.0 M), after which an aliquot of 5.0 mL of sodium hydroxide (1.0 M) was taken and titrated with hydrochloric acid (1.0 M) and phenolphthalein as an indicator [29]. The procedure was carried out in triplicate, estimating the average value in mg $C-CO_2$ $kg^{-1}$ of dry compost.

#### 2.4.2. Biological Indicators

Biological indicators were analyzed in triplicate, including those related to enzymatic, microbiological and ecophysiological activities in the composting systems.

#### Enzyme Indicators

The biological characterization consisted of the quantification, analysis and profiling of enzymatic activities related to the C, N and phosphorus (P) cycles. Specific dehydrogenase enzyme activities (DA) were determined using the INT (Iodophenyl-3-p-nitrophenyl-5-phenyltetrazolium) method [30]. Briefly, 1.0 g of compost was weighed and mixed with 1.5 mL of TRIS buffer and 2.0 mL of INT solution (9.88 mM). The samples were incubated at 40 °C for two hours. Afterward, an enzymatic extraction was performed using 10.0 mL of a mixture of ethanol and DMF (N,N-dimethylformamide) (1:1 v $v^{-1}$), homogenized for 20 min at 50 rpm. The mixture was then centrifuged at 2000 rpm for ten minutes in

a ZEIGEN CH90-1A centrifuge (Ningbo, China). The supernatant was quantified at a wavelength of 464 nm. Results were reported as nmol INTF kg$^{-1}$ of compost h$^{-1}$.

Urease activity (UA) was determined by adding 20.0 mL of borate buffer solution to 5.0 g of compost, plus 2.5 mL of freshly prepared urea solution (1:6 w v$^{-1}$). The samples were incubated for two hours at 37 °C. For the control sample, 2.5 mL of sterile distilled water was added instead of urea. Then, 6.0 mL of potassium chloride solution (7.46% w v$^{-1}$) was added to each test tube and shaken at 1500 rpm for 30 min. After that, the contents were gravity filtered using Whatman No. 2 filter paper (Solna, Sweden). Finally, 1.0 mL of filtrate was used to determine the N-NH$_4^+$ content by spectrophotometric method. The spectrophotometric analysis consisted of adding 9.0 mL of sterile distilled water, 5.0 mL of sodium salicylate-sodium hydroxide solution (1:1:1) and 2.0 mL of sodium dichloroisocyanurate solution (0.1% w v$^{-1}$) to the filtrate. The samples were left to stand for 30 min in the dark at room $T$, then the enzyme activity was measured at a wavelength of 690 nm. The results were expressed as nmol N-NH$_4^+$ kg$^{-1}$ compost h$^{-1}$ [31].

The overall activities of proteases, lipases and esterases were analyzed by fluorescein diacetate hydrolysis (FDA) [32]. FDA hydrolysis was determined by weighing 1.0 g of compost, to which 15.0 mL of phosphate buffer and 0.15 mL of FDA were added. The mixture was then stirred for one hour at 50 rpm. Subsequently, 2.0 mL of acetone was added and centrifuged at 4000 rpm at 4 °C for ten minutes. The supernatant obtained was used for the quantification of enzyme activities by spectrophotometry at a wavelength of 490 nm, reported as nmol of fluorescein kg$^{-1}$ of compost h$^{-1}$.

In addition to the evaluation of individual enzyme activities, an index called synthetic enzyme index (SEI) [5,33] was developed, reflecting the overall enzyme activity in the composting systems. The SEI included the enzymatic activities of DA, UA and FDA, reported in nmol kg$^{-1}$ compost h$^{-1}$. For the development of the SEI, Equation (1) was used:

$$SEI = \sum_{i=1}^{k} X_i \tag{1}$$

where $X_i$ is the concentration of the enzyme activity of DA, UA and FDA.

At the same time, the Shannon diversity index ($H'$) was also developed, which evaluates the enzymatic functional diversity in composting systems, where high values of the indicator ($H' > 4.0$) reflect a high metabolic capacity of the microorganisms present in the samples. The $H'$ indicator was developed using Equation (2):

$$H = -\sum_{i=1}^{k} (X_i * ln(X_i)) \tag{2}$$

where $X_i$ is the concentration of the enzyme activity of DA, UA and FDA.

In addition, the API ZYM$^{®}$ system was used as a complement to the enzyme analysis. The API ZYM$^{®}$ system included the testing of 19 specific enzyme activities in order to establish an enzyme profile over time in the composting systems [34]. The API ZYM$^{®}$ system consisted of a gallery of microcups containing chromogenic dehydrated substrates to determine enzyme activities belonging to the following enzyme families: (i) glycosyl hydrolases (α-Galactosidase, β-Galactosidase, β-Glucuronidase, α-Glucosidase, β-Glucosidase, N-Acetyl-β-glucosaminidase, α-Mannosidase and α-Fucosidase); (ii) proteases (Trypsin and α-Chymotrypsin); (iii) aminopeptidases (Leucine arylamidase, Valine arylamidase and Cystine arylamidase); (iv) esterases (Esterase, Esterase-lipase and Lipase); and (v) phosphatases (Alkaline phosphatase, Acid phosphatase and Naphthol-AS-BI-phosphohydrolase) [35]. For enzyme determinations, aqueous extracts from a compost:distilled water mixture (1:3 w v$^{-1}$) were used. The mixture was shaken at 650 rpm for 20 minutes. Subsequently, the samples were centrifuged at 2430 rpm for ten minutes at 25 °C. The supernatant was filtered using Whatman No. 2 filter paper (Solna, Sweden). Aliquots of 65.0 μL of the filtrate were added to each of the 20 microcups of the API ZYM$^{®}$ system and incubated

for four hours at 37 °C. After incubation, 30.0 µL of ZYM A and ZYM B reagents were added to each microculture and left to stand for five minutes [35]. The reactions generated color patterns, which were compared using the color codes and intensities established by the manufacturer and previously reported studies [5,35]. The intensity levels were: very high (Level 5, 40 nmol), high (Level 4, 30 nmol), medium (Level 3, 20 nmol), low (Level 2, 10 nmol), very low (Level 1, 5 nmol) and no intensity (Level 0, 0 nmol) [5].

Microbiological and Ecophysiological Indicators

During the dynamics of C and N mineralization, compost samples were also taken and microorganisms were extracted using Ringer's solution ($1:9$ w v$^{-1}$), which was composed of 8.2 g of sodium chloride, 4.18 g of potassium chloride, 3.32 g of calcium chloride, 1.9 g of monopotassium phosphate and 3.46 g of magnesium sulfate in 1.0 L of distilled water. The samples were shaken at 200 rpm for 20 min [34] and the obtained indicators were labeled as bacterial (BAC), fungi (FUN) and actinomycetes (ACT). In order to determine the BAC indicator, a 1:10 dilution with sterile saline (0.9% w v$^{-1}$) was performed, taking 0.1 mL aliquots and placing them in Petri dishes with BD Bioxon® nutrient agar solid medium [36]. The planting technique used was spread plate and the cultures were incubated at 37 °C for a period of 24–72 h.

The FUN indicator was determined following the methodology of Scheu and Parkinson [37], using BD Bioxon® malt extract agar solid medium. The Petri dishes were incubated at a *T* interval of 25 to 28 °C for seven days. Regarding the ACT indicator, the colonies were grown at 25 °C for 15 days according to the methodology described by Wellington and Toth [38]. For the calculation of the indicators BAC, FUN and ACT, the colonies developed on the aforementioned solid media were determined and reported as CFU g$^{-1}$ of dry compost in accordance with Zuberer [36].

To establish the evolution of microbial biomass during the mineralization dynamics, the microbial biomass C indicator (MBC) was quantified by the fumigation-extraction method [39], where samples of 20.0 g of compost were fumigated with water-free chloroform and incubated for 24 h at 25 °C [40]. Subsequently, the chloroform was removed from the samples and 100 mL of potassium sulfate (0.5 M), concentrated sulfuric acid and barium chloride (0.4% w v$^{-1}$) were added, after which the mixture was stirred for 40 min and filtered with Whatman No. 2 filter paper to obtain an extract. The extract was quantified spectrophotometrically at a wavelength of 600 nm, reporting the results in g $C_{mic}$ kg$^{-1}$ of compost.

As a complement to the evolution of the MBC indicator, the microbial biomass N indicator (MBN) was also determined, using the method established by Joergensen and Brookes [41]. Briefly, 0.6 mL of previously fumigated extract were taken, and 1.4 mL of citric acid buffer and 1.0 mL of ninhydrin agent were added. It was then placed in a water bath for 25 min. Afterward, 4.0 mL of ethanol:water solution (1:1 v v$^{-1}$) was added. The supernatant obtained was quantified at a wavelength of 570 nm and reported in g $N_{mic}$ kg$^{-1}$ of compost.

The metabolic quotient (q$CO_2$)—as an important biological indicator—was estimated at the beginning and at the end of the composting process. This quotient has been obtained by calculating the respiration rate of the biological phase (e.g., soils (C-$CO_2$)) divided by its MBC [42]. This indicator has also been used in the analysis of composts and other environmental matrices, in order to evaluate the physiological state of the biological phase, reported in g C-$CO_2$ kg$^{-1}$ of compost.

As an indicator of compost maturity, the analysis of the GI was performed during the composting process, following the methodology of Tiquia et al. [43]. The GI was based on the determination of primary germination. For this purpose, 10.0 g of compost were weighed and diluted in 100 mL of sterile distilled water; the mixture was stirred at 120 rpm for one hour and then filtered by gravity with Whatman No. 2 filter paper. Subsequently, 10.0 mL of the filtrate was added to 20 lettuce seeds of the species *Lactuca sativa*, incubating

them for five days at 22 °C in a LUZEREN DHP 952 incubator (Guadalajara, Jal, Mexico) and expressing the results as a percentage.

### 2.5. Statistical Analysis

The software R version 3.6.3 (R Development Core Team, Vienna, Austria) [44] was used for statistical analyses. The normality of the data was determined by the Shapiro-Wilk normality test with a significance level of $p \leq 0.05$. Statistical differences between the evaluated indicators were estimated by a two-way ANOVA with a Tukey's mean test with significance level $p \leq 0.05$ [45]. To establish the correlations between the indicators, a Pearson's product-moment correlation matrix was developed, considering a linear correlation $r^2 > \pm 0.6$ as a significant correlation [46]. A PCA was carried out, where a minimum data set (MDS) was established for the analyzed indicators. The PCA started with a normalization by natural logarithms ($y = ln(x)$) of the values of the assessed indicators. A Kaiser-Meyer-Olkin (KMO) data adequacy analysis [47] was then performed to test the suitability of the data for the PCA. The KMO analysis indicated the variance present in the data due to underlying factors, where high (KMO > 0.5) or low (KMO < 0.5) values indicated the suitability or unsuitability of the data for the PCA, respectively. The criterion of eigenvalue > 1 was also used to select the PCs [48]. Once the PCs were established, the indicators that presented a significant linear correlation with their PC ($r^2 \geq \pm 0.06$) were selected [46], having a commonality > 0.6 with their PC [49]. After that, a redundancy reduction process was carried out among the indicators related to their PC, under the following criteria and in order of importance: number of significant interactions > PC membership (PC1 > PC2 > ... > PCn) > correlation with their PC [50]. The quality results obtained by the developed CQIs were analyzed by a nonparametric Friedman's ANOVA, with subsequent Dunnett's test of difference of medians with Bonferroni adjustment, using a significance level of $p \leq 0.05$.

### 2.6. Development of Compost Quality Indexes (CQIs)

For the development of the indexes, three different methodologies were used: additive index ($CQI_a$), unified additive weight index ($CQI_w$) and nemoro index ($CQI_n$).

For the establishment of the $CQI_a$, Equation (3) was used [51]:

$$CQI_a = \frac{\sum_{i=1}^{n} S_i}{n} \tag{3}$$

where $S_i$ is the value of the scored indicator resulting from the redundancy reduction process and $n$ is the number of indicators included in the $CQI_a$.

For the establishment of the $CQI_w$, the methodology employed by Yu et al. [52] was followed, using the unified additive weight equation (Equation (4)) and the indicator scoring equations (Equations (5) and (6)) [52]. The unified additive weight equation was developed using the PC variability obtained in the process of developing the CQIs, which offers advantages over other techniques—such as fixed additive weight equation, expert opinion and linear additive indexes—and makes it one of the most widely adopted by the scientific community, allowing for comparison with other studies [33].

$$CQI_w = \sum_{i=1}^{n} W_i S_i \tag{4}$$

where $W_i$ is the proportion of PC variability to which the indicator is correlated, $S_i$ is the value of the scored indicator resulting from the redundancy reduction process obtained from the analysis of the compost samples.

Equation (5) was used to score the indicators whose role in the compost was considered as "the more the better" or "the less the better":

$$S_i = \frac{a}{1 + \left(\frac{X}{X_m}\right)^b} \tag{5}$$

where *a* is equal to the maximum value of the indicator, $X_m$ is the mean value of the indicator obtained from the analyses, *X* is the value of the indicator and *b* is the slope of the indicator score function ($-2.5$ and $2.5$ for indicators whose function was considered to be " the more the better" and "the less the better", respectively).

Equation (6) was used to score the indicators whose function in the compost was considered "optimal" and whose maximum or optimal value was 0.5 [53]:

$$S_i = \frac{1}{\left[1 + \left(\frac{B-L}{X-L}\right)^{2L(B+X-2L)}\right]} \tag{6}$$

where *B* is the value of the indicator whose slope is equal to 0.5, *L* is the lower limit value of the indicator and *X* is the value of the indicator.

For the establishment of the $CQI_n$, Equation (7) was used [51]:

$$CQI_n = \sqrt{\frac{P_{ave}^2 + P_{min}^2}{2} \times \frac{n-1}{2}} \tag{7}$$

where $P_{ave}^2$ and $P_{min}^2$ are the average and minimum value of the indicators resulting from the redundancy reduction process and *n* is the number of indicators included in the CQI.

The objective of the CQIs was to obtain a value between 0 and 1, thus establishing the quality of the compost, where 1 would represent maximum quality and 0 very poor quality.

## 3. Results and Discussion

### 3.1. Physicochemical Characterization of BS and BM

Table 1 shows the concentration of heavy metals in the BS and BM samples, as well as the maximum permissible limits according to the Mexican standard NOM-004-SEMARNAT-2002 [11] and the international rule EPA 503 [12]. The obtained results showed that the BS and BM samples were in the quality category for direct contact use. Therefore, it was decided not to include their analysis during the dynamics of C and N mineralization in the evaluated composting systems and in the development of the CQIs. Another reason for not including them is the fact that it has been reported that in the composting process, the bioavailability of heavy metals is reduced by the formation of trace metal elements [54], which is not an important variability factor to consider in the development of CQIs.

**Table 1.** Concentration of heavy metals in BS and BM.

| Heavy Metal (mg kg$^{-1}$) | Samples | | NOM-004-SEMARNAT-2002 and EPA 503 |
|---|---|---|---|
| | **BS** | **BM** | |
| As | 8.401 | 1.475 | 41 |
| Cd | 0.638 | <0.005 | 39 |
| Cr | 33.22 | 3.46 | 1200 |
| Cu | 123.00 | 19.00 | 1500 |
| Hg | <1.00 | <1.00 | 17 |
| Ni | <0.25 | <0.25 | 420 |
| Pb | 215.20 | 135.9 | 300 |
| Zn | 714.00 | 116.0 | 2800 |

All the results are presented on a dry-weight basis.

The values of the physicochemical indicators during the initial characterization of BS and BM are presented in Table 2. The *T* of BS was 42.3% higher than the one of BM; the difference was probably due to the post-stabilization of the sludge, increasing its *T* [5]. However, the initial values of BS and BM were found to be in the range of those reported by other authors [13,54]. Regarding M, there was a difference in values between the substrates (12.7%), being higher in BS; this difference could be mainly due to the stabilization process of BS [5] and to the characteristic composition of BM, as the latter has been reported to contain large amounts of organic residues and fibers resistant to degradation, thus reducing the moisture holding capacity [55,56]. The pH of BS was neutral, while the one of BM was alkaline. The EC of BS was 1.9 dS m$^{-1}$, 17.4% lower than that of BM (2.3 dS m$^{-1}$), which corresponds with the high availability of macro and micronutrients reported in similar studies of BS added with BM [6]. However, it should be taken into consideration that the addition of BM in excess or without prior characterization could lead to a possible accumulation of ions [56,57]. Nonetheless, the values obtained for the EC indicator were within those reported for composting processes, which can make the substrates be regarded as suitable [58].

**Table 2.** Initial values of physicochemical indicators for BS and BM.

| Indicators | BS | BM |
|:---:|:---:|:---:|
| *T* | 44.4 | 25.6 |
| M | 47.1 | 41.1 |
| pH | 7.2 | 9.9 |
| EC | 1.9 | 2.3 |
| BD | 0.09 | - - - |
| N-NO$_2^-$ | 0.014 | 0.019 |
| N-NO$_3^-$ | 0.474 | 3.103 |
| N-NH$_4^+$ | 0.006 | 0.000 |
| N$_{min}$ | 1.36 | 4.86 |
| NI | 0.0168 | 0.0001 |
| N-NH$_3^+$ | 11.2 | 20.9 |
| TN | 0.32 | 0.08 |
| SOC | 35.0 | 56.0 |
| TOC | 7.8 | 4.26 |

*T*, temperature (°C); M, moisture (%); pH, hydrogen potential; EC, electrical conductivity (dS m$^{-1}$); BD, bulk density (g cm$^{-3}$); N-NO$_2^-$, nitrite (mg N-NO$_2^-$ kg$^{-1}$ of BS or BM); N-NO$_3^-$, nitrate (mg N-NO$_3^-$ kg$^{-1}$ of BS or BM); N-NH$_4^+$, ammonium (mg N-NH$_4^+$ kg$^{-1}$ of BS or BM); NI, nitrification index (N-NH$_4^+$/N-NO$_3^-$); TN, total N (%); SOC, soluble organic C (mg SOC kg$^{-1}$ of BS or BM); TOC, total organic C (%).

BS presented four times more TN than BM, being a source of N to be exploited by the microbial biomass present during the mineralization of C and N sources. The difference in N content may be due to the nature of the physicochemical composition of the co-substrates that were used, e.g., BS had a higher concentration of mineralizable N—mainly N-NH$_4^+$—and a lower C/N ratio [2]. However, the SOC in BM was 37.5% higher compared to BS because BM by nature of its composition presented not only a higher amount of C compounds in general, but also heavier C molecular structures (fiber or more recalcitrant C compounds). Higher SOC values in BM could be a consequence of the degradation of complex C compounds by the microorganisms [54,56]. As for TOC concentration in BS, it was 45.4% higher than in BM. It is important to note that SOC and TOC values have been reported as reservoirs of C sources for the microbial biomass activity present in composting processes. High TOC values in composting systems increase SOC values as the metabolism of the microorganisms acts on the OM, converting complex organic fractions into more labile and easily degradable fractions, producing an increase in microbial activity [54]. Nevertheless, there was great variability in both BM and BS with respect to these indicators, mainly due to the sources of their collection, therefore, a previous characterization is necessary for their use in composting systems [59].

### 3.2. Dynamics of C and N Mineralization in Composting Systems

Physicochemical Indicators

Regarding the *T* indicator, there were significant differences ($p \leq 0.05$) between treatments with respect to time, presenting the following order: T1 > T3 > T2 > T4 with values 21.1, 28.0 and 57.3% lower—respectively—compared to T1 (Figure 1A). The range of *T* reached in the different treatments was between 45.2 and 19 °C, with an average of 24.6 °C. From the beginning of the mineralization dynamics, temperatures were mesophilic but close to thermophilic [58,60], decreasing in all the treatments up to day 15. Subsequently, there was a gradual increase in *T* until it reached a range between 20 and 30 °C in all the treatments on day 90, and there were no significant differences ($p > 0.05$) between them during this period. Finally, the *T* indicator reached a value of 19 °C after 120 days, a mesophilic temperature characteristic of the maturation stage of the composting process [13]. It is important to mention that a higher *T* value in T1—compared to the other treatments—could be due to the metabolism of a different microbial community, linked to a higher content of M [4].

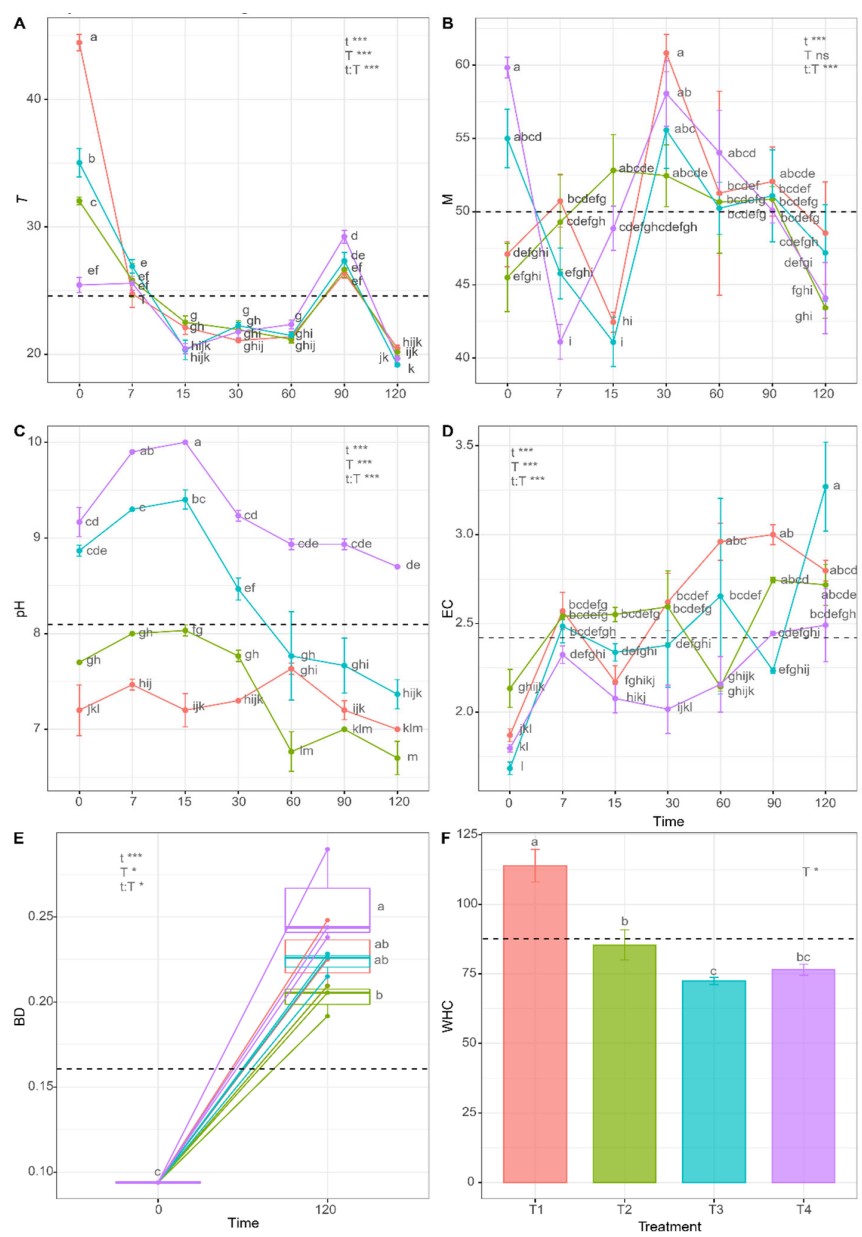

**Figure 1.** *Cont.*

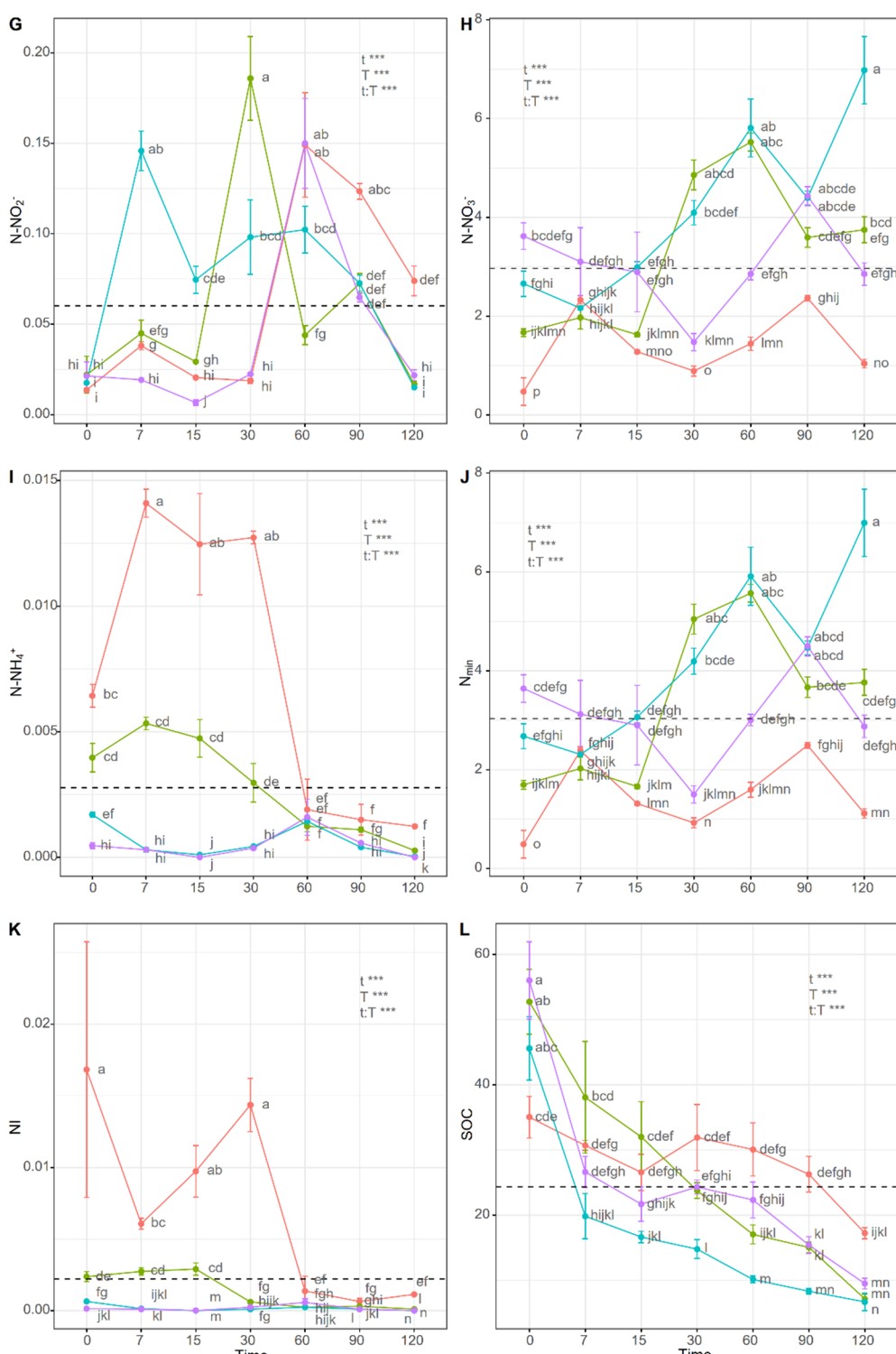

**Figure 1.** *Cont.*

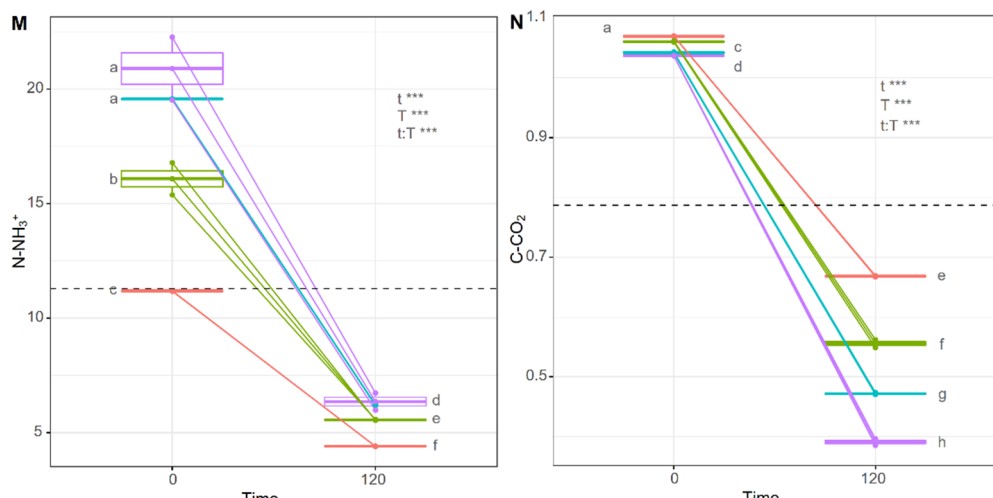

**Figure 1.** Two-way ANOVA of composting systems during C and N mineralization dynamics. Different letters indicate significant differences according to the two-way ANOVA with subsequent Tukey's test of means at a significance level $p \leq 0.05$. Time (days); T, treatment (T1, red line; T2, green line; T3, blue line; T4, purple line). (**A**) $T$, temperature (°C); (**B**) M, moisture (%); (**C**) pH, hydrogen potential; (**D**) EC, electrical conductivity (dS m$^{-1}$); (**E**) BD, bulk density (g cm$^{-3}$); (**F**) WHC, water holding capacity (%); (**G**) N-NO$_2^-$, nitrite (mg N-NO$_2^-$ kg$^{-1}$ dry compost); (**H**) N-NO$_3^-$, nitrate (mg N-NO$_3^-$ kg$^{-1}$ dry compost); (**I**) N-NH$_4^+$, ammonium (mg N-NH$_4^+$ kg$^{-1}$ dry compost); (**J**) N$_{min}$, mineralized N (mg N kg$^{-1}$ dry compost); (**K**) NI, nitrification index; (**L**) SOC, soluble organic C (mg SOC kg$^{-1}$ dry compost); (**M**) N-NH$_3^+$, ammonia (mg N-NH$_3^+$ kg$^{-1}$ dry compost); (**N**) C-CO$_2$, evolved C-dioxide (mg C-CO$_2$ kg$^{-1}$ dry compost). Vertical bars represent the standard deviation ($n$ = 5). Dotted line represents the mean of the indicator; ns, not significant; *, significant ($p \leq 0.05$); ***, highly significant ($p \leq 0.001$).

Similarly, the M indicator remained in a range of 39.8 to 62.2% in all the treatments, with an average of 50% and no significant differences between them ($p > 0.05$). This could be due to the recurrent addition of water to the composting systems in order to avoid evapotranspiration or desiccation of the systems, therefore, the M content remained in the abovementioned range (Figure 1B). It has been reported that M contents in the range of 34 to 43% allow for reaching thermophilic stages in the composting processes, a phenomenon observed in the present study during the first days of C and N mineralization dynamics in the treatments T1, T3 and T4 [58].

With regard to the pH indicator (Figure 1C), at the beginning of the mineralization dynamics there were significant differences ($p \leq 0.05$) between treatments, having a tendency towards alkalinity, which showed a positive relationship with the C/N ratio of the runs. Alkalinity in the treatments could be linked to the addition of BM (T4 > T3 > T2 > T1), being that the initial pH values were 28.2, 23.6 and 7.6% higher than T1, respectively. In general, the pH indicator presented values in the range of 6.5 to 10, with an average of 8.1. An increase in the pH indicator was observed, reaching a maximum on day 15 in the treatments, except for T1, whose maximum value (7.7) was reached until day 60. The dynamics of the pH indicator over time could also be linked to the volatilization of N-NH$_3^+$ and the mineralization of N compounds with production of N-NO$_3^-$, as a consequence of the types of N present in the co-substrates and the activity of the microorganisms present in the composting systems [1,13]. Subsequently, a generalized decrease in pH was observed in all the treatments, being T1, T2 and T3 in a neutral range at the end, whereas T4 was in an alkaline range, with the following order of the pH indicator being observed: T4 > T3 > T1 > T2. A decrease in pH at the end of the processes could be a result of the degradation of OM due to the production of organic acids, which would neutralize the salts produced by OM mineralization (Figure 1C) [13].

In context, the EC indicator showed an increase at the beginning of the mineralization dynamics, with significant differences ($p \leq 0.05$) between treatments T1, T2 and T4, with respect to T3. By day 120 there were no significant differences between them ($p > 0.05$) and the following order for the EC indicator was established: T3 = T1 = T2 = T4 (Figure 1D). It should be noted that, from day 7 onward, all the treatments were considered to have salt concentrations in the range of 1.65 to 3.55 dS m$^{-1}$, with an average of 2.42 dS m$^{-1}$ (Figure 1D). An increase in the EC indicator toward the end of the mineralization dynamics could be due to the mineralization process of the various C and N compounds or molecules present in the systems [61]. Additionally, it has been reported that increased EC is linked to increased N-NO$_3^-$ and N-NH$_4^+$ and decreased SOC [13,62]. It is also important to note that, in composting systems, EC concentrations can affect plant growth and the development of microbial communities. However, other studies have reported higher values compared to the ones obtained in this research, suggesting a previous characterization of the utilized co-substrates and the final use or destination of these composts [5].

As for the BD indicator, there were significant differences ($p \leq 0.05$) between treatments with respect to time. The BD indicator showed a tendency to increase, with values in the range of 0.09 to 0.29 g cm$^{-3}$, and an average of 0.16 g cm$^{-3}$. The treatments presented the following BD order: T4 > T1 = T3 > T2, with significant differences ($p \leq 0.05$) only between treatments T4 and T2 (Figure 1E). The increase in the BD indicator may be due to the compaction of the compost as a result of OM degradation, which in turn is linked to the action of microorganisms present in the composting systems. Both OM and BD have been considered to be indicators of the maturity and stability of composts [54]. On the other hand, a lower BD value at the beginning of the C and N mineralization dynamics could not only have allowed for microbial growth and proliferation, but also for the adequate aeration of the system, facilitating the aerobic degradation of OM and preventing the development of anaerobic zones [14].

Regarding the WHC indicator, at the end of the mineralization dynamics (day 120) there were significant differences ($p \leq 0.05$) between treatments, presenting values in the range of 71.3 to 119.8%, with an average of 87.1%. It was observed that the higher the amount of BM in the treatment, the lower the WHC (T1 > T2 = T4 = T3) (Figure 1F). A lower WHC in the BM treatments could be due to the high content of fibrous material that was difficult to degrade, which increased the porosity of the system, preventing water retention, with a tendency to further desiccation. This decrease in WHC may also affect the degradation of OM and therefore influence the porosity in less compacted composting systems [63].

The C-CO$_2$ indicator showed values in the range of 0.39 to 1.07 mg C-CO$_2$ kg$^{-1}$ of dry compost, with an average of 0.79 mg C-CO$_2$ kg$^{-1}$ of dry compost, indicating significant differences ($p \leq 0.05$) between treatments. The treatments presented the following order of C-CO$_2$ concentration: T1 > T2 > T3 > T4 (Figure 1N). At the beginning of the mineralization dynamics T2, T3 and T4 had values 0.8, 2.6 and 3.1% lower than T1, however, at the end their values were 16.9, 29.3 and 41.5% lower than T1, respectively. The behavior of the C-CO$_2$ indicator reflected an increase in microbial activity at the beginning of the mineralization dynamics, being higher for T1 due to the presence of a higher fraction of labile N compounds, while for the remaining treatments such fractions may have been influenced or compromised by the incorporation of BM [64]. Nonetheless, the decrease in the concentration of C-CO$_2$ in the different treatments—except for T1—at the end of the dynamics (day 120) has been stipulated as an indication of the maturity stage in the composting systems, due to the degradation of OM [54]. Likewise, it could be mentioned that the addition of BM had a positive effect on the degradation of OM, observing—in comparison to T1—a decrease in the C-CO$_2$ indicator in the different treatments at the end of the mineralization dynamics (ranging from 16.9 to 41.5%). Moreover, adequate aeration of the composting systems possibly contributed to the development of more diverse microbial communities with the capacity to mineralize various C and N sources from the co-substrates used in this study [14].

The SOC indicator showed significant differences between treatments ($p \leq 0.05$) only at the end of the mineralization dynamics, compared to T1 (Figure 1L). Values of SOC were observed in a range from 5.19 to 61.99 mg SOC kg$^{-1}$ of dry compost, with an average of 24.34 mg SOC kg$^{-1}$ of dry compost, presenting the following order according to their concentration at the end of the mineralization dynamics: T1 > T4 = T2 = T3. Therefore, the SOC values were 44.9, 58.6 and 61.2% lower than T1—respectively—toward the end of the mineralization dynamics (Figure 1L). This coincides with other authors [65] who reported an increase in C concentrations in the treatments, followed by a decrease, due to microbial activity through the use of C compounds by the microorganisms and an increase in N. It is important to note that all the treatments showed a decreasing trend from the beginning of the mineralization dynamics. A higher degradation of C compounds represented by the SOC indicator and a lower production of C-CO$_2$ has been reported as an indication of the maturation of the composting systems, as well as the fixation of C in possibly more recalcitrant compounds such as fulvic acids [54].

Furthermore, the N-NH$_3^+$ indicator showed significant differences between treatments with respect to time ($p \leq 0.05$), with values ranging from 4.40 to 22.67 mg N-NH$_3^+$ kg$^{-1}$ of dry compost and an average of 11.28 mg N-NH$_3^+$ kg$^{-1}$ of dry compost. In order of concentration, the N-NH$_3^+$ indicator was observed as follows: T4 > T3 > T2 > T1, with values at the beginning of the mineralization dynamics 86.8, 75.0 and 43.8% higher than T1, respectively (Figure 1M). At the end of the mineralization dynamics, the concentrations remained in the same order, being 44.4, 40.9 and 26.2% higher than T1, respectively (Figure 1M). Higher N-NH$_3^+$ volatilization was observed as a function of BM application rate in the treatments. A higher volatilization showed by this indicator at the beginning could be a result of higher microbial activity due to the addition of C and labile N compounds—as witnessed by the concentration of SOC in treatments T4, T3 and T2 (Figure 1L)—and an increase in the concentration of C-CO$_2$ (Figure 1N), as well as the N requirements of the microorganisms and the tendency to denitrification processes [54]. A higher concentration of N-NH$_3^+$ in the BM-added treatments could be a consequence of alkaline pH values that affected the oxidation process of N-NH$_4^+$, thus producing its volatilization and loss in the form of N-NH$_3^+$ [13,66]. Consequently, conditions for adequate conversion of N-NH$_4^+$ to oxidized N compounds—such as N-NO$_2^-$ and N-NO$_3^-$—were compromised until days 7 (T2), 15 (T3) and 30 (T4) [67]. The above coincides with the study by Awasthi et al. [68], who obtained higher N-NH$_3^+$ production in the thickener-added treatments than in the controls (biosolid composting).

On the N-NH$_4^+$ indicator, significant differences ($p \leq 0.05$) were observed between treatments with respect to time. The N-NH$_4^+$ indicator presented values in the range of 0 to 0.15 mg N-NH$_4^+$ kg$^{-1}$ of dry compost, with an average of 0.03 mg N-NH$_4^+$ kg$^{-1}$ of dry compost, having the following concentration order at the beginning of the dynamics: T1 = T2 > T3 = T4, where values were 38.3, 73.4 and 92.8% lower than T1, respectively (Figure 1I). However, in treatments T1 and T2, N-NH$_4^+$ concentrations increased, reaching their maximum concentration on day 7. Treatments T3 and T4 showed an opposite trend to T1 and T2, reaching their minimum concentration on day 15 (Figure 1I). After day 7, treatments T1 and T2 showed a decrease, whereas T3 and T4—after day 15—showed an increase in the concentration of N-NH$_4^+$. By day 60, there were no significant differences ($p > 0.05$) between treatments. Subsequently, all the treatments showed a decrease in their concentration of N-NH$_4^+$ up to day 120, in the following order of concentration: T1 > T2 > T3 > T4 (Figure 1I). Likewise, at the end of the mineralization dynamics, values were 78.6, 96.1 and 99.5% lower than T1, respectively. The concentration of the N-NH$_4^+$ indicator showed a tendency to decrease as a function of the doses of BM application in the various treatments. This could be due to the fact that T1—as it did not contain BM—had a higher labile fraction of N compounds, while the other treatments had less labile fractions to be degraded by microorganisms and also had a higher initial C/N ratio [58]. This can also be seen in Figure 1K, where high values of the indicator NI were observed in T1, presumably with a strong tendency for a denitrification process

to occur [54]. At the same time, it has been reported that high $C\text{-}CO_2$ concentration (Figure 1N) and alkaline pH conditions (Figure 1C) could produce inhibitory effects on the OM mineralization process under saline conditions [13,69]. Given the above, it could be established that the addition of BM in the treatments could favor N conversion processes via denitrification and presumably N volatilization processes (Figure 1M). At the end of the mineralization dynamics, low concentrations of the $N\text{-}NH_4^+$ indicator could be observed in all the treatments, confirming the maturation of the composting process. The difference in $N\text{-}NH_4^+$ concentration between T1 and the other treatments at the beginning of the mineralization dynamics could be due to a contribution of high values in the C/N ratio (>30), which limited the concentration of N available for the microorganisms, thus causing that in T3 and T4 the microorganisms could not fix the available N in their cellular structure, provoking its decrease via volatilization [15]. The tendency to decrease the concentration of the $N\text{-}NH_4^+$ indicator in composting processes has been reported by other authors [13,64] whose composting systems showed an increase in $N\text{-}NH_4^+$ concentration during the first few days, followed by a decrease until the end of the composting process.

On the other hand, the $N\text{-}NO_2^-$ indicator showed concentrations without variability, being constant for T2, T1 and T4 until days 15 and 30, respectively, whereas T3 presented its maximum concentration on day 7, with significant differences between treatments up to day 90 and the end of the mineralization dynamics ($p \leq 0.05$). At the end of the mineralization dynamics, the following concentration order was observed: T1 > T4 = T3 = T2, with values 70.6, 77.1 and 79.5% lower than T1, respectively (Figure 1G). In general, the $N\text{-}NO_2^-$ indicator showed values in the range of 0.006 to 0.208 mg $N\text{-}NO_2^-$ $kg^{-1}$ of dry compost, with an average of 0.06 mg $N\text{-}NO_2^-$ $kg^{-1}$ of dry compost. Higher concentrations of the indicator $N\text{-}NO_2^-$ have been considered as an indication of oxidative mineralization processes of N compounds, being an intermediate compound of $N\text{-}NH_4^+$ oxidation to obtain $N\text{-}NO_3^-$. This is consistent with the decrease of the $N\text{-}NH_4^+$ indicator (Figure 1I) at the end of the mineralization dynamics, as well as the fact that anaerobic conditions are presumably not present due to the manual-mechanical mixing process regularly carried out in the composting systems.

Regarding the $N\text{-}NO_3^-$ indicator, there were significant differences between treatments with respect to time ($p \leq 0.05$). The $N\text{-}NO_3^-$ indicator presented values in the range of 0.23 to 7.75 mg $N\text{-}NO_3^-$ $kg^{-1}$ of dry compost, with an average of 2.97 mg $N\text{-}NO_3^-$ $kg^{-1}$ of dry compost. At the beginning of the mineralization dynamics, the following order of concentration of the $N\text{-}NO_3^-$ indicator was observed: T4 = T3 = T2 > T1, with values 664.5, 461.2 and 252.3% higher than those of T1, respectively. In the treatments added with BM, there was a decrease in the concentration of $N\text{-}NO_3^-$ from day 7 (T3), 15 (T2) and 30 (T4), subsequently increasing the concentration of $N\text{-}NO_3^-$, while for T1 it remained unchanged until the end of the mineralization dynamics. At the end of the dynamics, the following order of concentration of the $N\text{-}NO_3^-$ indicator was observed in the treatments: T3 > T2 = T4 > T1 (Figure 1H), showing values 559.8, 259.8 and 173.9% higher—respectively—compared to T1. The increase of this indicator was attributed to the mineralization and oxidation processes of N compounds ($N\text{-}NH_4^+$, $N\text{-}NH_3^+$ and $N\text{-}NO_2^-$), since it has been observed, for example, that the presence of various organic and inorganic co-substrates in BS promotes N mineralization by microorganisms [70]. Other factors involved in the mineralization of N compounds are $T$ and pH, with a low concentration of $N\text{-}NO_3^-$ being observed in the initial stages of the composting process (thermophilic phase and alkaline pH), increasing its concentration at a mesophilic $T$ and neutral pH (Figure 1A,C) [13]. This corresponds with what was observed on the $N\text{-}NO_3$ indicator, having low concentrations at the beginning of the mineralization dynamics and an increase toward the end (day 120). The addition of BM as a compost co-substrate presumably improved aeration conditions, allowing for the proliferation of nitrifying microorganisms. This was related to the UA indicator (see below), which showed a maximum value on day 15 of the mineralization dynamics (Figure 2B) presumably stimulated by the addition of

N in the treatments when adding BM [69]. An increase in the concentration of N-NO$_3^-$ is considered as the beginning of maturation in the various composting systems [63].

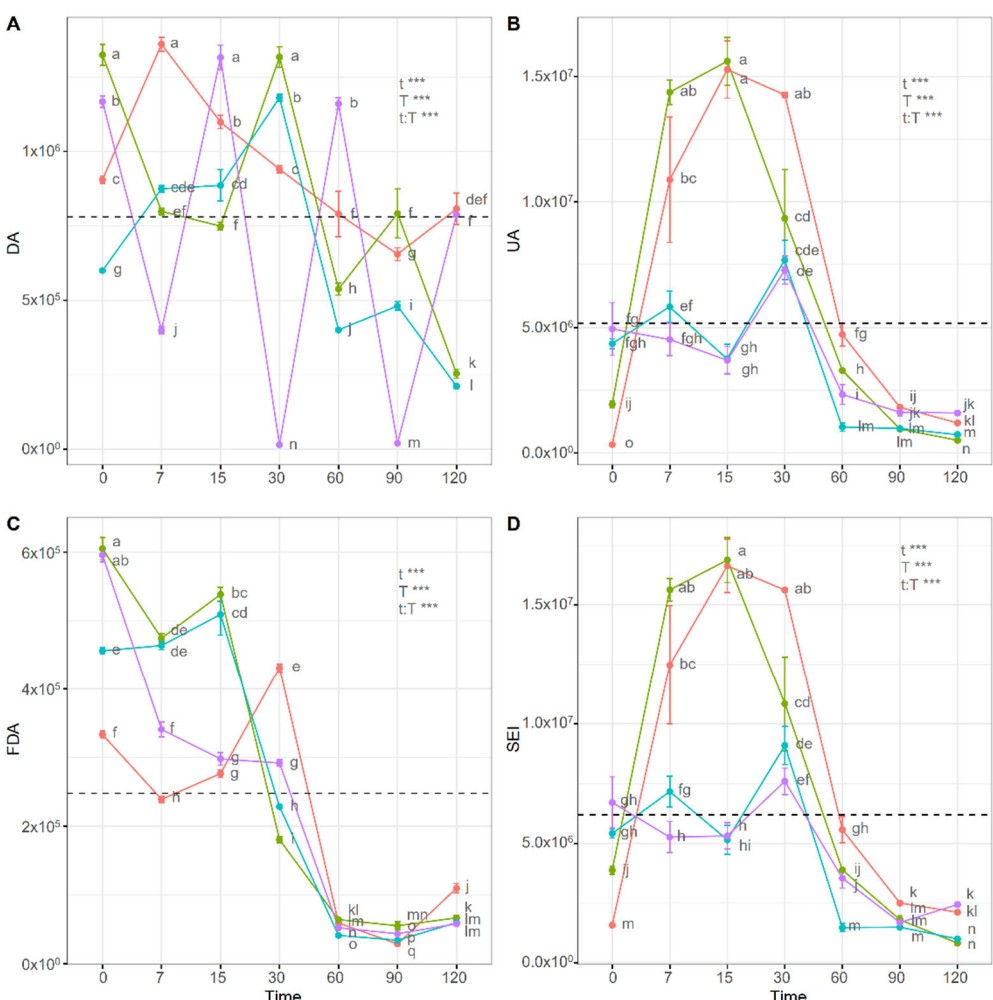

**Figure 2.** Two-way ANOVA of enzyme indicators of composting systems. Different letters indicate significant differences according to the two-way ANOVA with subsequent Tukey's test for means at a significance level $p \leq 0.05$. Vertical bars represent the standard deviation ($n = 5$). (**A**) DA, dehydrogenase activity (nmol INTF kg$^{-1}$ compost h$^{-1}$); (**B**) UA, urease activity (nmol N-NH$_4^+$ kg$^{-1}$ compost h$^{-1}$); (**C**) FDA, fluorescein diacetate (nmol fluorescein kg$^{-1}$ compost h$^{-1}$); (**D**) SEI, synthetic enzyme index (nmol kg$^{-1}$ compost h$^{-1}$). Dotted line represents the mean of the indicator; ***, highly significant ($p \leq 0.001$).

At the same time, the N$_{min}$ indicator showed a similar trend to the N-NO$_3^-$ indicator, decreasing at the beginning of the mineralization dynamics and then increasing until the end of the composting process, with values in the range of 0.24 to 7.76 mg N kg$^{-1}$ of dry compost, and an average of 3.03 mg N kg$^{-1}$ of dry compost, showing significant differences between the treatments and T1 ($p \leq 0.05$). At the beginning of the mineralization dynamics, the treatments presented the following concentration order for the N$_{min}$ indicator: T4 = T3 = T2 > T1, with values 664.5, 461.2 and 252.3% higher than T1, respectively. At the end of the mineralization dynamics, the various treatments presented the following concentration order T3 > T2 = T4 > T1 (Figure 1J), with values 526.1, 237.1 and 157.4% higher—respectively—compared to T1. This is in agreement with what was mentioned about the N-NH$_4^+$ and N-NO$_3^-$ indicators, which showed variations due to nitrification processes during the mineralization dynamics. It has been previously observed that a decrease in N-NH$_4^+$ concentration in composting systems under aerobic conditions is due

to its conversion into compounds such as $N-NO_2^-$ and, subsequently, $N-NO_3^-$ [13]. Thus, the increasing trend of the $N_{min}$ indicator supports the theory that the addition of BM as a co-substrate promoted the mineralization of nitrogenous compounds present in the composting systems mainly after day 30. The tendency of $N_{min}$ to decrease and subsequently to increase, is more similar to the one presented by the $N-NO_3^-$ indicator, possibly due to the fact that the mineralization of N compounds under adequate oxygenation conditions directs the N cycle in the composting system to a greater extent after day 30, avoiding the loss of N through $N-NH_3^+$ [64].

In reference to the NI indicator, there were significant differences between treatments with respect to time ($p \leq 0.05$), presenting values in the range of 0 to 0.026, with a mean of 0.02. The order of NI indicator concentration in the treatments at the beginning of the mineralization dynamics was as follows: T1 > T2 > T3 > T4, showing values 85.9, 96.2 and 99.2% lower than T1, respectively (Figure 1K). Specifically, a decrease of the indicator was observed in the BM treatments, with the exception of T2, which reached its maximum value on day 15, decreasing thereafter. The treatments presented significant differences until the end of the dynamics ($p \leq 0.05$), showing the following order: T1 > T2 > T3 = T4 (Figure 1K), with values 94.1, 99.4 and 99.8% lower—respectively—compared with T1. The high values in T1 and T2 possibly promoted the volatilization and fixation of N observed through the indicators $N-NH_3^+$, ACT and FUN, as well as the alkaline pH conditions in the composting systems [13]. This is supported by the fact that, in the first 30 days of the mineralization dynamics, no significant increase ($p > 0.05$) in the $N-NO_3^-$ indicator was observed in the treatments described above. For this reason, the addition of BM in the treatments allowed for the oxidative mineralization of organic N compounds from day 30, during the mesophilic stage [66]. Figure 1H,J confirm the increase in $N-NO_3^-$ and $N_{min}$ indicators, respectively. The measurement of indicators related to the N cycle has been implemented to measure the maturity and quality of the products of composting systems, since ammonification processes are intensified at the beginning of the process, while nitrification processes are intensified at the end, with the stabilization and maturation of the compost [63,71,72].

### 3.3. Biological Characterization

#### 3.3.1. Enzymatic Indicators

Enzyme activities are closely related to the development and growth of microbial communities in composting systems, as well as to the utilization of substrates and nutrients. Figure 2 shows the results for the enzyme indicators DA, UA, FDA and SEI.

For the DA indicator, the treatments showed significant differences with respect to time ($p \leq 0.05$), with values ranging from $1.47 \times 10^4$ to $1.39 \times 10^6$ nmol INTF $kg^{-1}$ of compost $h^{-1}$, with an average of $7.8 \times 10^5$ nmol INTF $kg^{-1}$ of compost $h^{-1}$. The order of concentration of the DA indicator at the beginning of the mineralization dynamics was: T2 > T4 > T1 > T3, with values 46.0 and 29.0% higher for T2 and T4, and 34.0% lower for T3, compared to T1. Treatments T2 and T4 showed a decrease, contrary to T1 and T3, which increased on days 7 and 30, respectively, decreasing afterward. At the end of the mineralization dynamics, the treatments presented the following order of concentration of the DA indicator: T1 = T4 > T2 > T3, with values 2.45, 68.53 and 73.78% lower than T1, respectively (Figure 2A). The decrease observed in T2 and T3 could be due to the presence of less labile C compounds, contrary to T1 and T4, where the latter—containing BM—presumably presented conditions conducive to the proliferation of microorganisms, with an increase in their enzymatic activity [55]. At the same time, the decrease in the concentration of the DA indicator at the end of the dynamics is an indication of the stability and maturation of the composting system, supported by the decrease in pH, SOC and $C-CO_2$, as well as the increase in mineralization, resulting in a higher concentration of the $N-NO_3^-$ indicator [63].

In reference to the UA indicator, there were significant differences ($p \leq 0.05$) between treatments with respect to time, presenting values in the range of $3.25 \times 10^5$ to

$1.67 \times 10^7$ nmol N-NH$_4^+$ kg$^{-1}$ of compost h$^{-1}$, with an average of $5.17 \times 10^6$ nmol N-NH$_4^+$ kg$^{-1}$ of compost h$^{-1}$. At the beginning of the C and N mineralization dynamics, UA presented the following order of concentration in the treatments: T4 = T3 > T2 > T1, observing values 1345.8, 1173.1 and 468.1% higher than T1, respectively. In T1 and T2, an increase in UA concentration was observed until day 15, followed by a decrease until the end of the mineralization dynamics. T3 and T4 showed a decrease in UA concentration until day 15, then an increase in concentration until day 30, followed by a decrease until the end of the mineralization dynamics. On day 120, UA presented the following concentration order in the treatments: T4 = T1 > T3 > T2 (Figure 2B), with values 32.76% higher in T4, and 39.27% and 45.74% lower in T3 and T2—respectively—compared with T1. The significant increase ($p \leq 0.05$) for T1 and T2 was consistent with the composition or mixture of the established substrates, as T1 and T2 had a higher proportion of BS, which had a higher concentration of labile nitrogenous compounds such as N-NH$_4^+$ [5,63]. This was also in agreement with the increase in the N-NH$_4^+$ concentration observed in T1 and T2 in Figure 1I. The decrease in UA concentration in T3 and T4 was linked to a higher C/N ratio (44 and 54, respectively) suggesting a lower access to labile nitrogenous compounds present in the OM, causing a slow growth of microorganisms at the beginning of the C and N mineralization dynamics [15]. At the end of the dynamics (day 120) a decrease in UA concentration was observed in all the treatments, which was stated as a signal or biological event linked to the stabilization and maturation process of the composting systems.

Regarding the FDA indicator, the treatments showed significant differences ($p \leq 0.05$), with values ranging from $2.77 \times 10^4$ to $6.24 \times 10^5$ nmol of fluorescein kg$^{-1}$ of compost h$^{-1}$, with an average of $2.48 \times 10^5$ nmol of fluorescein kg$^{-1}$ of compost h$^{-1}$. The treatments presented the following order for the FDA indicator concentrations at the beginning of the mineralization dynamics: T2 = T4 > T3 > T1, showing values 81.7, 78.9 and 37.0% higher than T1, respectively. The FDA values in T3 and T1 showed an increase in FDA concentration until day 15 and 30, respectively, then a decrease. At the end of the mineralization dynamics, the treatments showed significant differences ($p \leq 0.05$) with respect to the FDA indicator, observing the following concentration order: T1 > T2 > T3 = T4 (Figure 2C). The estimated FDA percentages of T2, T3 and T4 were 39.27, 45.74 and 47.25% lower than T1, respectively. The difference in FDA concentrations between the BM treatments and T1 could be due to the fact that an increase in enzyme activity in composting processes has been linked to the addition of OM, which would increase access to labile C and N compounds for the present microorganisms; this coincides with the concentrations shown by the SOC and C-CO$_2$ indicators (Figure 1L,N). The decrease in the overall FDA concentration could be due to the action of extracellular protease enzymes, which act on the enzymes secreted by the microorganisms, causing a decrease in the overall enzyme activity toward the end of the mineralization dynamics [5]. Another factor that could contribute to the decrease in the overall enzyme activity is the increase of BD, causing a lower access of microorganisms to the OM, leading to a lower microbial activity and a decrease in the enzyme concentration (Figure 2A) [14].

The SEI was developed in this study as an indicator to summarize the overall enzymatic activity in the composting systems. The SEI presented significant differences ($p \leq 0.05$) between treatments with respect to time, with values in the range of $7.65 \times 10^5$ to $1.80 \times 10^7$ nmol kg$^{-1}$ of compost h$^{-1}$, and an average of $6.20 \times 10^6$ nmol kg$^{-1}$ of compost h$^{-1}$. The trends and order of concentration in the treatments for this indicator followed the same behavior as the UA indicator (Figure 2D). At the beginning of the mineralization dynamics, the following concentration order was observed on the SEI indicator: T4 = T3 > T2 > T1, with values 308.99, 230.57 and 137.65% higher than T1, respectively. In the same way as the UA indicator, the concentration of SEI decreased over time, where the following order of concentrations was observed in the treatments at the end of the dynamics: T4 = T1 > T3 = T2, with values 14.3% higher in T4, and 52.69 and 60.93% lower in T3 and T2—respectively—compared to T1. The overall enzyme activity represented by the SEI indicator is mostly influenced by the UA indicator and the degradation processes

of N compounds. This is supported by the indicators $N-NO_3^-$, $N-NH_4^+$, $N_{min}$ and NI (Figure 1H–K) because this indicator was related to the N cycle within the composting systems and because it was a limiting element for microbial growth due to the high C/N ratio in the BM-added treatments [15].

In the same context, the API ZYM® system made it possible to evaluate and complement the dynamics of the enzyme activity profiles of the composting systems. Nineteen enzymes related to C, N and P cycling were analyzed across treatments and time (Figure 3). The concentrations of enzyme activities in the following order were observed in the treatments: T1 > T2 > T3 > T4. With regard to the activity of the enzyme families, they presented the following order: phosphatases > glycoacyl hydrolases > peptidases > lipase esterases > aminopeptidases. In addition to the abovementioned, individual enzyme activities were observed in the following order: Alkaline phosphomonoesterase = α-Fucosidase > Acid phosphomonoesterase = Trypsin = α-Chymotrypsin > Phosphohydrolase = Lipase esterase > N-acetyl-β-glucosaminidase > Cystine arylamidase > Esterase = β-Glucosidase > α-Glucosidase = β-Glucosidase > β-Galactosidase > Lipase = Valine arylamidase > Leucine arylamidase > α-Mannosidase > α-Galactosidase. With respect to time, enzyme activities generally decreased as the days of the mineralization dynamics elapsed. A higher activity of the phosphatase enzyme family was related to the low availability of P in the composting systems, forcing the microorganisms to secrete enzymes (alkaline and acid phosphomonoesterase) for the incorporation of P into their structure (membrane and energy accumulation) [33,34]. Moreover, increased enzymatic activity of alkaline phosphatase was observed. The activity of this enzyme has been reported under neutral to slightly alkaline pH conditions, which coincides with the conditions presented by T1 and T2 (Figure 1C) [63]. Phosphatase activity generally increased as a function of the BM concentration added to the studied treatments, possibly because BM has been reported to provide a source of P to composting systems (Figure 3). In the same context, the activity of the glycosyl hydrolase enzyme family is related to the C cycle in composting systems and its acquisition by the present microorganisms. A higher activity of this family at the beginning of the mineralization dynamics could be a consequence of higher availability of C compounds due to the addition of BM [34,55]. On the other hand, a decrease in the activity of the glycosyl hydrolase enzyme family at the end of the mineralization dynamics could be explained by a decrease in OM in the composting systems and developed treatments, which is also related to a decrease in microbial activity and the SOC indicator, leading to a decrease in $C-CO_2$. The activity of the peptidase family is related to the degradation of proteins to amino acids with the release of organic acids, which could lead to a decrease in the pH of the composting systems. In this study, the peptidase family showed higher activity in T1 and decreased over time as a function of the addition of BM [34,73]. The esterase-lipase and aminopeptidase enzyme families showed the lowest activity because the former is related to the degradation of water-soluble C compounds (SOC) (ester bonds and organic acids) and the latter to the degradation of amino acids from protein mineralization. At the beginning of the mineralization dynamics, the esterase-lipase and aminopeptidase enzyme families had a higher concentration of substrates (labile C and N compounds) on which to act, so it was suggested that the microbial community in the systems did not require a higher production of extracellular enzymes.

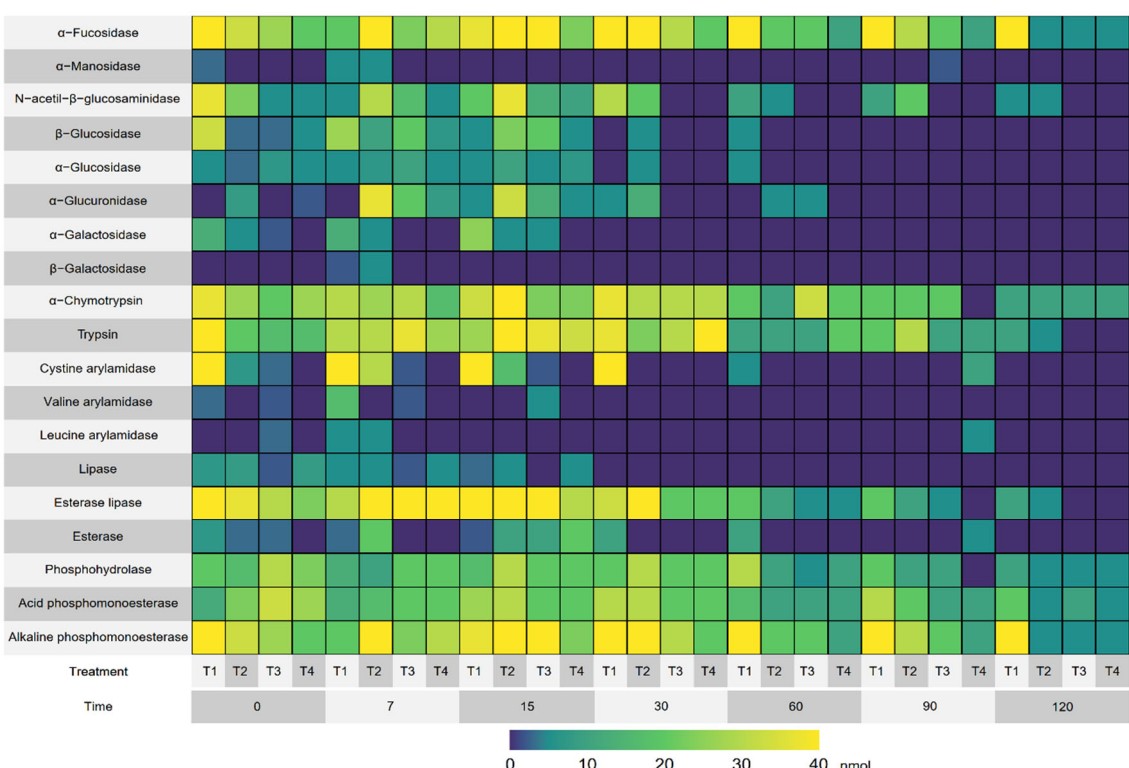

**Figure 3.** Enzyme profile of composting systems.

### 3.3.2. Microbiological and Ecophysiological Indicators

The concentration of specific microbial groups in composting systems has been an indicator of the stability of a compost, as well as the utilization of the various nutrient pools [16]. Figure 4 shows the dynamics over time of the BAC, ACT, FUN and *H'* indicators.

Regarding the BAC indicator, there were no significant differences ($p > 0.05$) between treatments with respect to time, presenting values in the range of $6.78 \times 10^5$ to $1.59 \times 10^8$ CFU $g^{-1}$ of dry compost, with an average of $3.43 \times 10^7$ CFU $g^{-1}$ of dry compost. At the end of the mineralization dynamics, the following concentration order was observed for the BAC indicator: T3 = T1 = T2 = T4, with values 16.1 and 4.8% lower in T2 and T4—respectively—and 35.2% higher in T3, compared to T1. In general, the BAC indicator decreased over time in all the treatments (Figure 4A). The decrease in BAC concentration was possibly due to the decrease in C and labile N sources, which was supported by the SOC and N-NH$_4^+$ indicators (Figure 1L,I) [73], and confirmed by the DA enzyme activity, related to viable cells and enzyme activity in the composting systems (Figure 2A). The decrease in the BAC indicator and related enzyme activity in all the treatments was considered to be an indication of the steady state of the composting systems [63].

The ACT indicator presented significant differences ($p \leq 0.05$) between treatments from day 15 of the mineralization dynamics. The observed values were in the range of $1.49 \times 10^5$ to $1.43 \times 10^7$ CFU $g^{-1}$ of dry compost, with an average of $2.63 \times 10^6$ CFU $g^{-1}$ of dry compost (Figure 4B). On day 120 of the mineralization dynamics, the ACT indicator in the treatments presented the following order: T1 > T2 = T3 > T4, with values 68.3, 74.9 and 90.0% lower than T1, respectively. The concentration of actinomycetes increased in the first days of the mineralization dynamics, with maximum concentrations on days 30 (T3), 60 (T2), 90 (T4) and 120 (T1). This group of microorganisms has been reported to be involved in the degradation of long-chain C compounds (cellulose and lignin) [70], therefore, the increase in their concentration at the beginning of the dynamics would be related to the degradation of long-chain C compounds and the release of shorter C compounds (SOC) and C-CO$_2$. The major impact factor in ACT growth is the C content [70], consequently, the decrease

in ACT at the end of the mineralization dynamics was related to the decrease in SOC and C-CO$_2$ (Figure 1L,N).

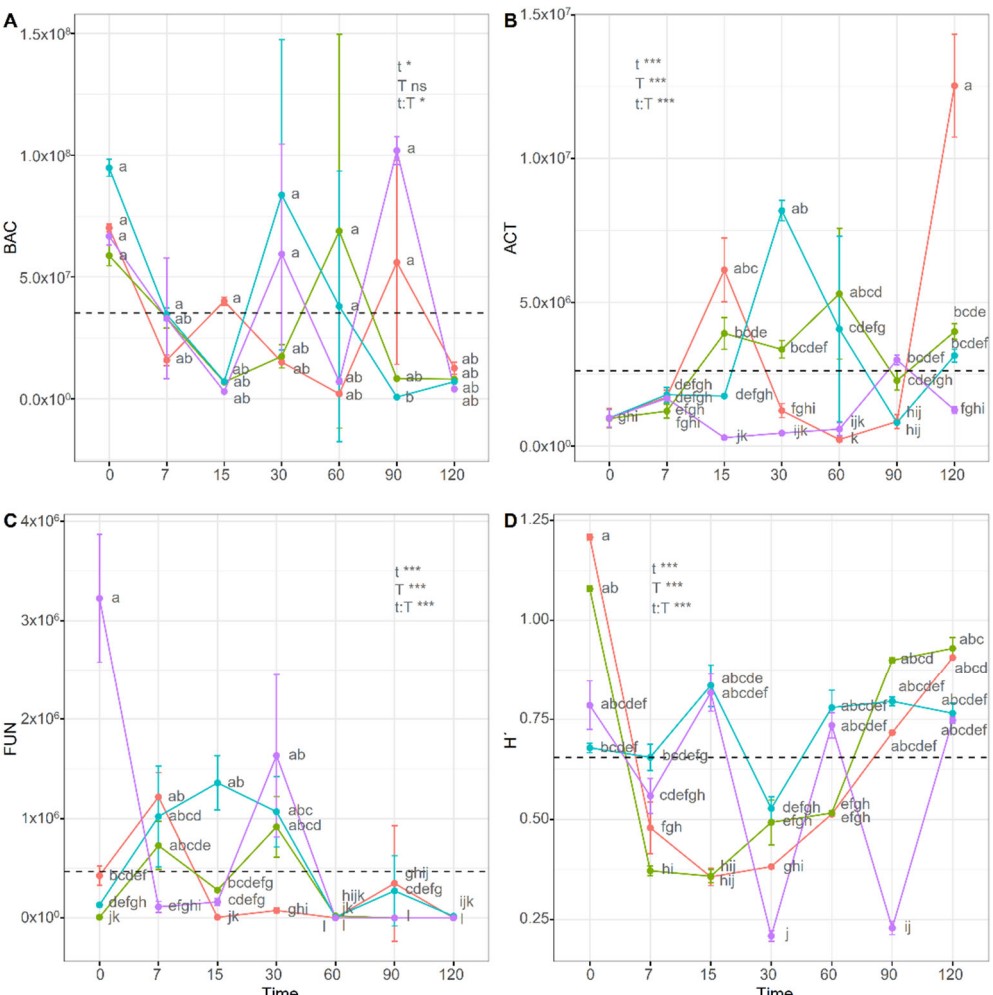

**Figure 4.** Two-way ANOVA for Shannon index and microbiological indicators of composting systems. Different letters indicate significant differences according to the two-way ANOVA with subsequent Tukey's test of means at a significance level $p \leq 0.05$. Vertical bars represent the standard deviation ($n = 5$). (**A**) BAC, bacteria (CFU g$^{-1}$ dry compost); (**B**) ACT, actinomycetes (CFU g$^{-1}$ dry compost); (**C**) FUN, fungi (CFU g$^{-1}$ dry compost); (**D**) $H'$, Shannon index. Dotted line represents the mean of the indicator; ns, not significant; *, significant ($p \leq 0.05$); ***, highly significant ($p \leq 0.001$).

As for the FUN indicator, it showed significant differences between treatments with respect to time ($p \leq 0.05$), with values in the range of 0 to $3.87 \times 10^6$ CFU g$^{-1}$ of dry compost, and an average of $4.65 \times 10^5$ CFU g$^{-1}$ of dry compost. The treatments presented the following concentration order for the FUN indicator at the beginning of the mineralization dynamics: T4 > T1 = T3 > T2, with values 660.3% higher in T4, and 69.5% and 98.5% lower in T3 and T2—respectively—compared to T1. Treatments T1, T2 and T3 had an increase during the first days of the mineralization dynamics, reaching their maximum concentrations on days 7, 15 and 30, respectively. Subsequently, there was a decrease in this indicator over time. In the case of T4, there was a decrease in the concentration of FUN at the beginning of the mineralization dynamics, then an increase on day 30, followed by a decrease until the end of the dynamics (Figure 4C). The increases over time in the FUN indicator in the different treatments could be due to the presence of cellulose and lignin compounds in the added BM, which were decreasing in concentration toward the end of the mineralization dynamics (day 120) [34]. On the other hand, a decrease in the concentration

of FUN could be interpreted as a state of maturation of the composting system toward the end of the mineralization dynamics [63].

In addition, the $H'$ indicator has been reported to be a reflection of the enzymatic diversity in the environmental systems. This indicator did not show significant differences ($p > 0.05$) between the treatments with respect to time (Figure 4D), having values in the range of 0.20 to 1.21. The average value was 0.66 and was considered as low diversity ($H' < 4.0$) for an environmental system [5]. In all the studied treatments, there was a decrease in enzyme diversity in the first days of the mineralization dynamics, followed by an increase in enzyme diversity until the end of the mineralization dynamics (Figure 4D). The $H'$ indicator presented the following order of: T2 = T1 = T3 = T4, with values 2.56% higher, and 15.6 and 17.5% lower than T1, respectively. A decrease in enzyme diversity during the first 15 (T1 and T2) and 30 days (T3 and T4) could be due to the decrease in the concentration of microorganisms present in the composting systems, and it could linked to the dynamics of the indicators BAC, ACT and FUN. Furthermore, the decrease in easily accessible or labile C and N compounds may have affected the expression of enzymes in the microorganisms (Figure 3), i.e., as time went by, the activity of the various enzymes decreased and even reached a level at which their activity could not be detected. This was also consistent with the concentration of other indicators such as SOC, $N\text{-}NH_3{}^-$, $N\text{-}NH_4{}^+$ and $C\text{-}CO_2$. In summary, based on the $H'$ values, it was established that the microbial communities present in the composting systems over time and in their treatments were not affected by the addition of the utilized substratum, nor by the time period of the mineralization dynamics, which could mean that the microorganisms were well adapted to the physical and chemical conditions of the system [6].

Regarding the ecophysiological indicators, the results obtained are shown in Figure 5. The MBC indicator showed significant differences ($p \leq 0.05$) between treatments with respect to time, with values in the range of 75.31 to 1716.13 g $C_{mic}$ $kg^{-1}$ of compost, and an average of 535.68 g $C_{mic}$ $kg^{-1}$ of compost. MBC generally decreased from the beginning to the end (day 120) of the mineralization dynamics in all the treatments. However, on day 15, an increase in MBC was recorded for all the treatments, although it decreased thereafter. At the end of the dynamics, the composting systems showed an MBC concentration in the following order: T2 = T4 > T3 > T1 (Figure 5A), with values 381.7, 281.2 and 69.9% higher than T1, respectively. Being that the MBC indicator is related to the number of microorganisms present in the composting systems, a decrease in this indicator was linked to a decrease in the BAC and FUN indicators in this study. At the same time, the MBC indicator is closely related to the enzymatic activities in their ability to degrade OM present in the composting systems [68], therefore, a decrease in MBC indicated a lower concentration of OM, which in turn was reflected in a decrease in the SOC, $C\text{-}CO_2$, DA, UA and FDA indicators [5].

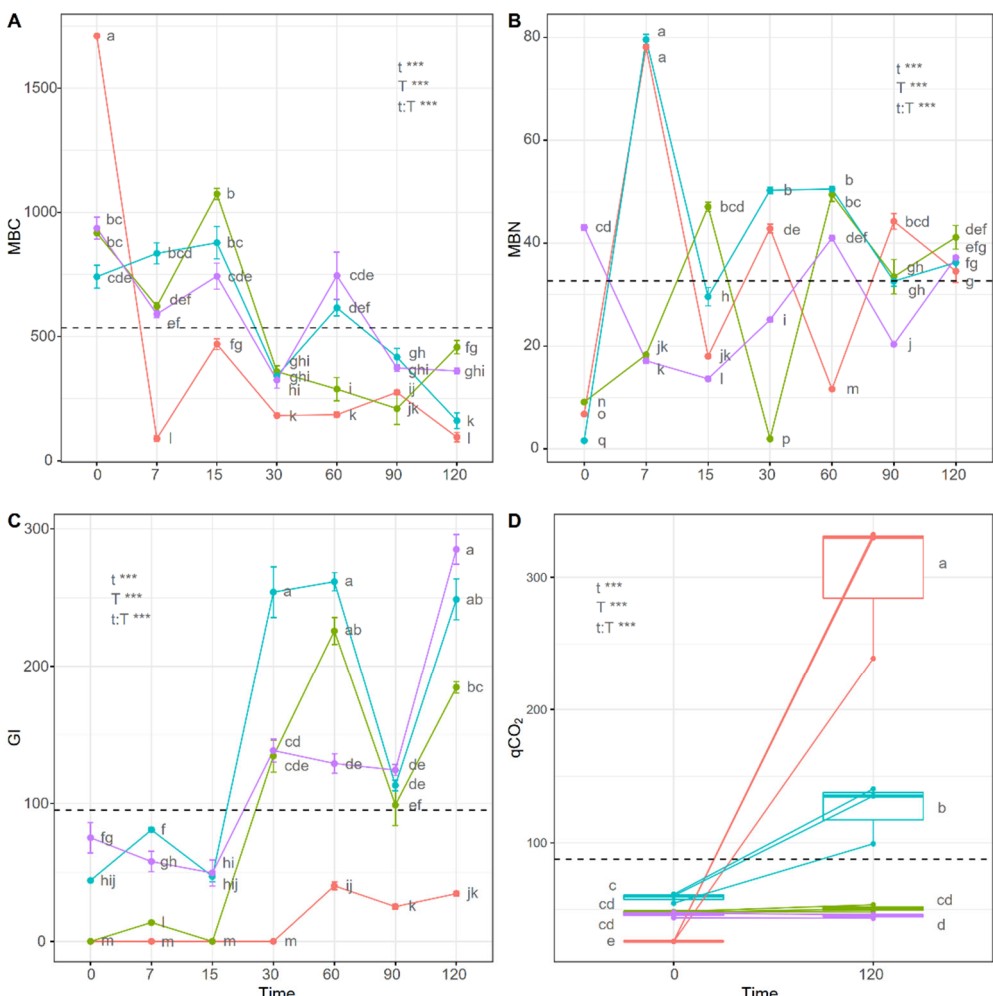

**Figure 5.** Two-way ANOVA for ecophysiological indicators of composting systems. Different letters indicate significant differences according to the two-way ANOVA with subsequent Tukey's test of means at a significance level $p \leq 0.05$. Vertical bars represent the standard deviation ($n = 5$). (**A**) MBC, microbial biomass C (g $C_{mic}$ $kg^{-1}$ compost); (**B**) MBN, microbial biomass N (g $N_{mic}$ $kg^{-1}$ compost); (**C**) GI, germination index (%); (**D**) $qCO_2$, metabolic quotient (g $C$-$CO_2$ $kg^{-1}$ compost). Dotted line represents the mean of the indicator; ***, highly significant ($p \leq 0.001$).

As for the MBN indicator, the treatments showed significant differences ($p \leq 0.05$) throughout the mineralization process, contrary to what was observed at the end of the process, where there were no significant differences ($p > 0.05$). At the end of the mineralization dynamics, the treatments showed the following order of concentration: T2 = T4 = T3 = T1, with values 19.1, 7.7 and 4.9% higher than T1, respectively. The MBN indicator presented values in the range of 1.39 to 80.38 g $N_{mic}$ $kg^{-1}$ of compost, with an average of 32.66 g $N_{mic}$ $kg^{-1}$ of compost. This indicator showed a slight tendency to increase in the composting systems during the mineralization dynamics and presented its maximum concentration values on day 7 in T1 and T3, while in T2 and T4 the maximum concentration was reached until day 60 (Figure 5B). The increase in the concentration of MBN was declared to be a process of biological fixation of the N present in the composting systems. This can be related to the behavior of the $N$-$NO_3^-$ and $N_{min}$ indicators, which increased at the end of the mineralization dynamics. However, the trend of the MBN indicator in the present study contrasts with what has been reported in other studies, in which an increase of the indicator is established as the composting process progresses [68].

Likewise, the GI indicator—which has been related to the phytotoxicity of the composts —presented significant differences between treatments with respect to time ($p \leq 0.05$),

showing values in the range of 0 to 295.91%, with a mean of 95.25%. The GI indicator showed a tendency to increase over time, with the treatments presenting the following order: T4 > T3 > T2 > T1 (Figure 5C), having values 722.9, 618.1 and 431.9% higher than T1, respectively. From the beginning of the mineralization dynamics, it could be observed that T3 and T4 presented better conditions with respect to phytotoxic compounds. From day 30 onward, all the treatments except T1 presented GI values higher than the ones recommended for composts (GI = 80%) [13]. Lower values in the early stages of composting could be due to the release of toxic substances in the process of decomposition of OM; as time passes, phytotoxic substances are eliminated or degraded by microorganisms [16]. The phytotoxicity in T1 and T2 at the beginning of the mineralization dynamics could be attributed to the fact that—as they contained a higher proportion of BS—phytotoxic substances were released in higher concentrations through the degradation of OM by the microorganisms [16]. The opposite process occurred in T3 and T4 due to the addition of a higher dose of BM, which allowed for better aeration of the system and at the same time added a higher concentration of microbial communities that facilitated the elimination of phytotoxic substances from the composting system. High values of the GI indicator (GI > 80%) made it possible to establish the state of maturity and quality of the composts in the different treatments used in this study, besides, values of GI > 100% are considered to be phytonutritious, which supports the obtained composting process [16].

Within the same context of analysis of ecophysiological indicators, the $qCO_2$ indicator showed significant differences ($p \leq 0.05$) between treatments with respect to time, having values in the range of 25.96 to 332.21 g $C\text{-}CO_2$ $kg^{-1}$ of compost, with an average of 87.52 g $C\text{-}CO_2$ $kg^{-1}$ of compost (Figure 5D). This indicator has been related to stress conditions and nutrient uptake by microbial communities in environmental systems [5,34]. The trend of the $qCO_2$ indicator for T1 and T3 was to increase over time, while for the other treatments it remained unchanged. At the end of the mineralization dynamics, the $qCO_2$ indicator presented the following order: T1 > T3 > T2 = T4 (Figure 5D), with values 56.4, 83.1 and 85.0% lower than T1, respectively. Higher values of the $qCO_2$ indicator in treatments T2, T3 and T4 could be related to the addition of OM to the composting systems, a behavior described in other investigations [5,34]. A higher value in T1 compared to the other treatments at the end of the mineralization dynamics could be a consequence of higher stress due to the low access of microorganisms to labile C and N compounds [5], contrary to treatments T2 and T4, which were added with OM. This OM provided better nutritional conditions for the composting systems, allowing for the maintenance of microbial activity through a higher concentration of nutrients during the mineralization dynamics.

*3.4. PCA*

In the Pearson's correlation matrix (Figure 6), it could be observed that the indicators $N\text{-}NO_3^-$, $N\text{-}NH_4^+$, $N_{min}$, NI, TOC, GI, and SEI showed significant correlations with at least one other indicator ($r^2 \geq \pm 0.6$). The remaining indicators did not show significant correlations with any other indicator. It is important to highlight that the addition of OM promoted an increase in the concentration of indicators such as NI (0.63) and SEI (0.64). The degradation of OM increased the phytotoxicity of the system, probably due to the release of toxic compounds when the microorganisms degraded the OM (GI = −0.62). On the other hand, mineralization processes of N compounds ($N\text{-}NO_3^-$ and $N_{min}$) decreased phytotoxicity by promoting the maturation and quality of the compost (GI = 0.73).

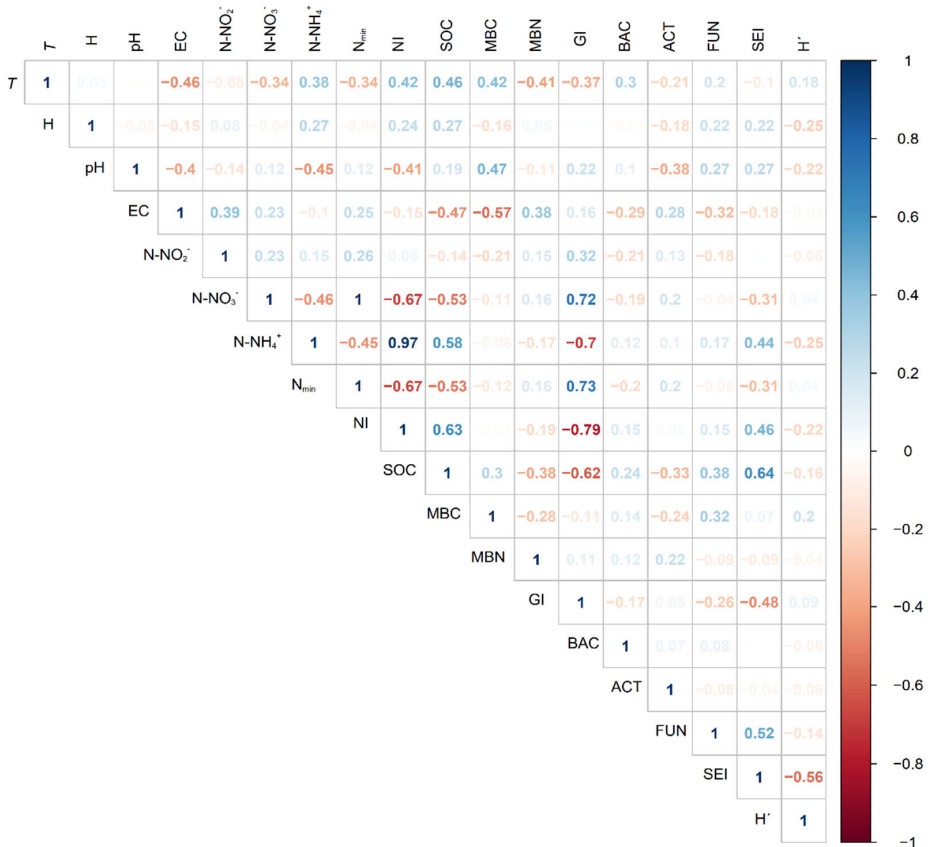

**Figure 6.** Pearson's product-moment correlation matrix. Positive correlation in blue, negative in red, no correlation not shown.

As a result of the PCA, two PCs were obtained, which fulfilled the criterion of eigenvalue > 1. The selected PCs accounted for 83.4% of the variability of the indicators analyzed during the mineralization dynamics. The variability was distributed as shown in Figure 7.

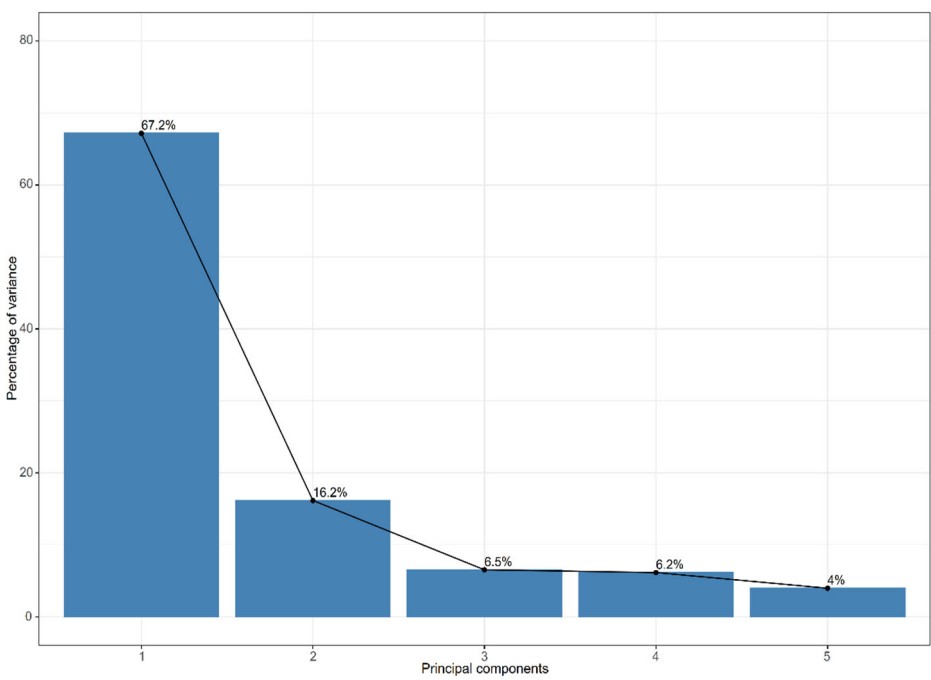

**Figure 7.** Percentage of variability of PCs.

The analyzed indicators showed significant correlations with their respective PCs. PC1 showed significant correlations with all the indicators, whereas PC2 only showed significant correlations with the SEI indicator (Figures 8 and 9).

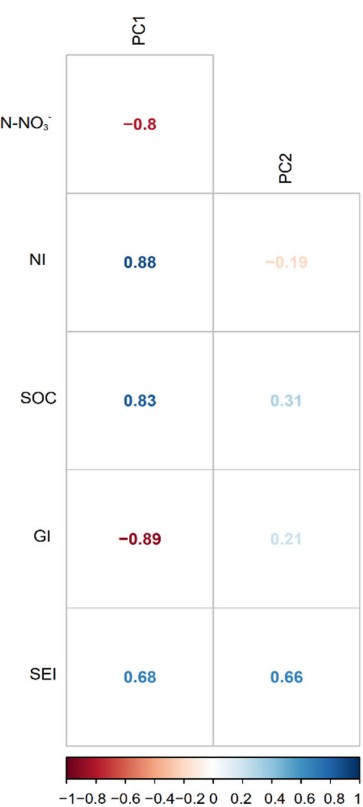

**Figure 8.** Pearson's correlation matrix between indicators and PCs.

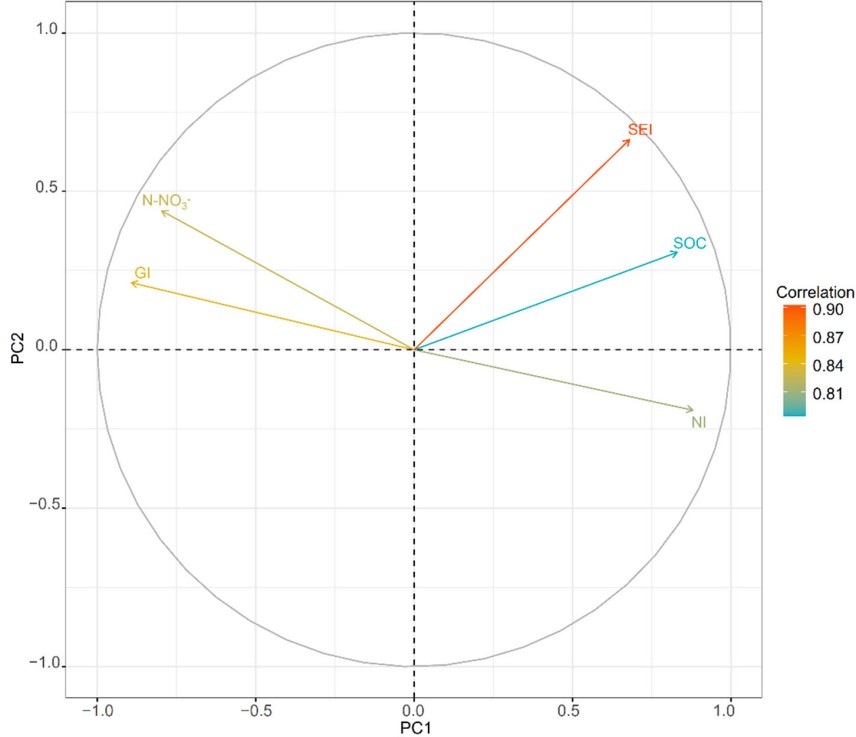

**Figure 9.** Bi-graphic indicators and PCs.

*3.5. Establishment of CQIs*

Due to their relevant results, the NI and SEI indicators were established as those with the highest relationship to compost quality [16]. These indicators were transformed using the scoring equation (Equation (5)), with the indicators being considered as having the functions of "the less the better" and "the more the better", respectively, in the composting systems. The NI indicator and its selection as a compost quality indicator coincided with what has been reported by Peña et al. [15] and by Meena et al. [16], whose studies established—through a PCA—indicators related to N cycling and phytotoxicity as the most related to compost quality. Saldarriaga et al. [14], through a PCA as well, concluded that moisture and respirometry (indicator related to enzymatic activity) are the factors that presented the closest relationship to the maturation, stability and quality of composts. The aforementioned studies presented a methodology similar to the one followed in the present study. However, this study seeks to contribute with a well-defined scale to relate the values of the selected indicators to the quality of the compost through the use of CQIs. Therefore, the indicators obtained and their applicability in the monitoring of composting systems highlight the importance of establishing the biosolids application dose with respect to their N concentration and their biological functionality—enzymatic activity. Previous studies carried out by the research group related to the application of biosolids to soils, were already incapable of monitoring indicators related to the N cycle and the diverse microbial activities of the soil [5,8,10]. Finally, the obtained CQIs are shown below:

$$\text{CQI}_a = \frac{S_{NI} + S_{SEI}}{2} \tag{8}$$

$$\text{CQI}_w = (0.672 \times S_{NI}) + (0.672 \times S_{SEI}) \tag{9}$$

$$\text{CQI}_n = \left( \sqrt{\frac{P^2_{ave_{NI}} + P^2_{min_{NI}}}{2}} \times \frac{1}{2} \right) + \left( \sqrt{\frac{P^2_{ave_{SEI}} + P^2_{min_{SEI}}}{2}} \times \frac{1}{2} \right) \tag{10}$$

The $\text{CQI}_w$ index classified the treatments into two groups: the highest quality, consisting of T2, T3 and T4, and the lowest quality, consisting of T1 (Figure 10), presenting an average value considered as high quality ($\text{CQI}_w = 0.62$), coinciding with what was obtained through the GI indicator, considered as a parameter of the maturity and quality of the compost products (Figure 5C). The $\text{CQI}_a$ index could not classify the treatments in the same way as $\text{CQI}_w$, obtaining a higher quality group that comprised treatments T3 and T4, an intermediate quality group for T2 and a lower quality group for T1 (Figure 11), presenting average value considered to be of moderate quality ($\text{CQI}_a = 0.56$). The $\text{CQI}_n$ presented an inverse tendency to the two previous ones, being T1 and T2 in the highest quality group, T3 in the intermediate quality group and T4 in the lowest quality group (Figure 12), presenting an average value considered as low quality ($\text{QCI}_n = 0.30$). The behavior of the different CQIs that were developed could be explained by the fact that the $\text{CQI}_a$ and $\text{CQI}_n$ indexes established an equal weight for the selected indicators—which is not completely true—and therefore did not allow for a proper classification of the quality of the different composting systems and treatments used in this study. This is contrary to what was obtained by the $\text{CQI}_w$, which—using a statistical technique—established the proportional weights of each indicator based on the variability of the selected PCs, thus facilitating a classification according to the quality of the composting systems in relation to the utilized treatments.

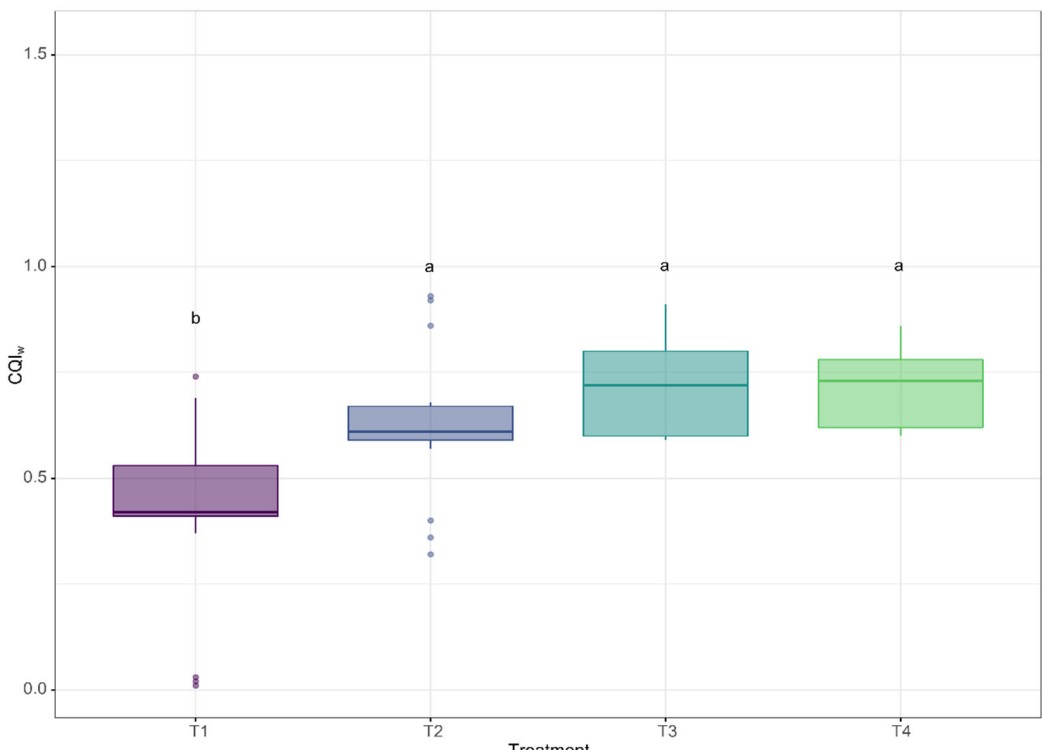

**Figure 10.** Friedman's analysis of compost quality using the $CQI_w$ index. Different letters indicate significant differences according to Dunnett's mean test with Bonferroni adjustment ($p \leq 0.05$).

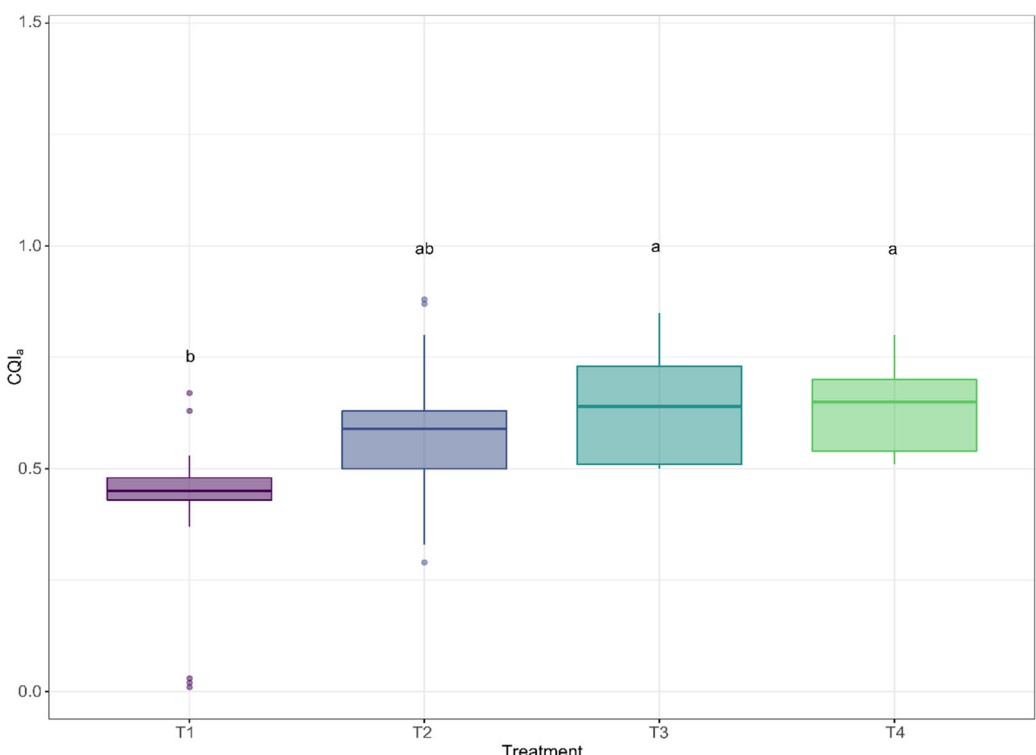

**Figure 11.** Friedman's analysis of compost quality using the $CQI_a$ index. Different letters indicate significant differences according to Dunnett's mean test with Bonferroni adjustment ($p \leq 0.05$).

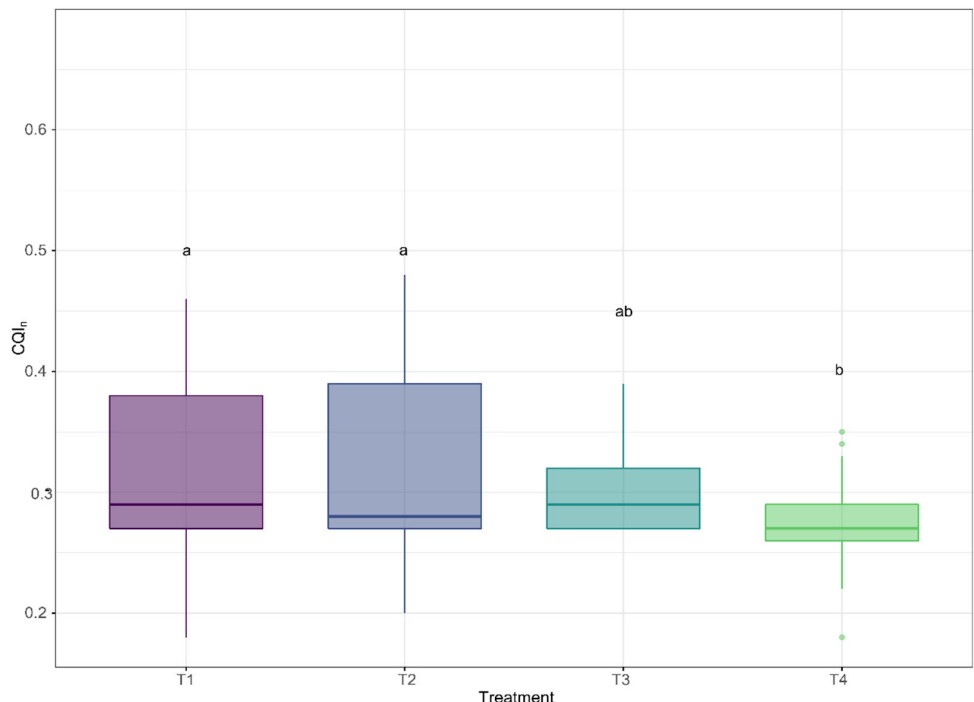

**Figure 12.** Friedman's analysis of compost quality using the $CQI_n$ index. Different letters indicate significant differences according to Dunnett's mean test with Bonferroni adjustment ($p \leq 0.05$).

In order to numerically compare the various CQIs developed in this study, linear correlation equations were established. Figure 13 shows that all the comparisons between the developed indexes were significant ($p \leq 0.05$), however, the highest correlation coefficient was presented in the comparison between $CQI_a$ and $CQI_w$ $\left(R^2 = 0.98\right)$ (Figure 13A).

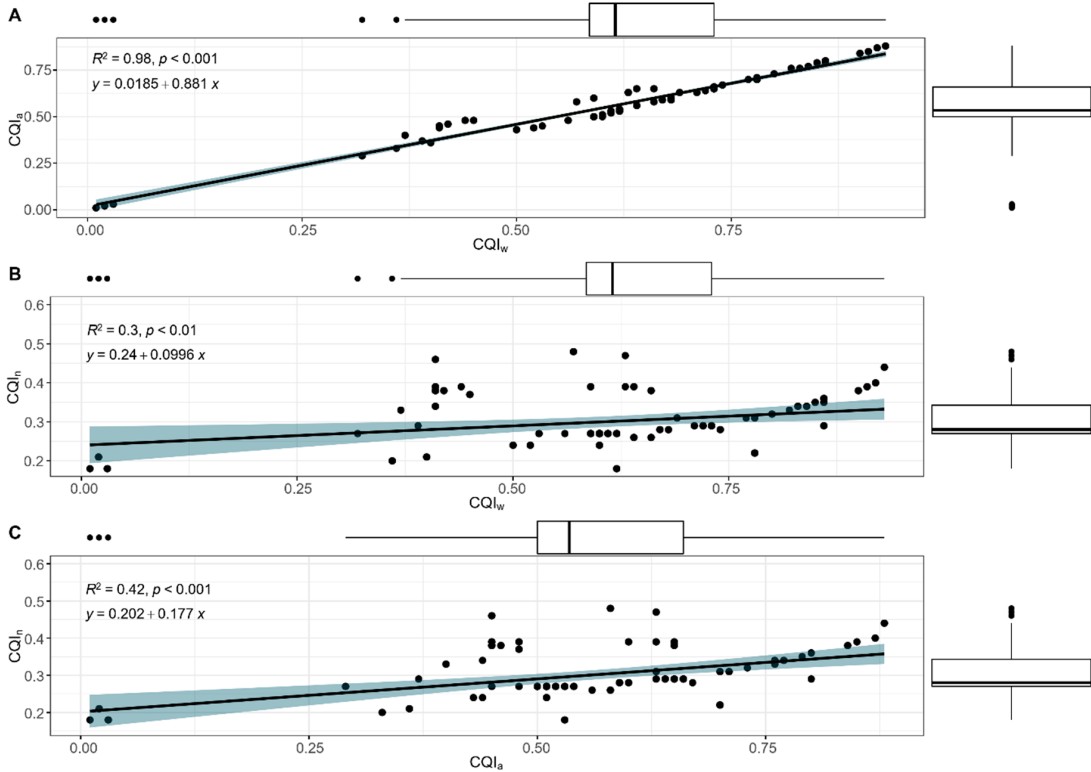

**Figure 13.** Linear correlation between developed CQIs; gray area represents the 95% CI (**A–C**).

*3.6. Applications, Challenges and Perspectives*

The developed $CQI_w$ is a tool that allows to measure in a fast, simple and easy way to interpret the product quality of BS composting systems, which gives clues of the most related indicators regarding the quality of the compost and the processes involved. This tool can be applied to similar systems around the world, reducing costs and evaluation periods, allowing for systematic monitoring. The challenges to be overcome in this type of research are based on the temporal variation of BS production, which must be taken into account in subsequent studies, in order to refine the CQIs developed. The perspectives for this study are the evaluation of the application of BS composts with different qualities obtained from the monitoring based on the developed $CQI_w$, in soils with low fertility, degraded or contaminated, with the purpose of observing the effect caused by the treatments.

## 4. Conclusions

The treatments with added BM generally improved the conditions of transformation and assimilation of C and N sources in the various composting systems, allowing for better aeration and higher addition of OM. The treatments used in this study promoted the loss of N via volatilization. The final products obtained from the composting systems showed conditions of maturity, stability and quality, reflected in low values of SOC and N compounds and high values of GI. Moreover, the PCA methodology used in this study allowed for the selection of the indicators that were most related to compost quality (NI and SEI), indicators selected in previous investigations as important to monitor in composting processes. The $CQI_w$ index presented the best performance in the classification of quality for the different utilized treatments. The $CQI_w$ developed stands out as an easy-to-interpret tool for measuring the quality of BS composts with BM and similar systems at the national and international level, reducing the number of indicators analyzed and therefore the difficulty and cost of monitoring.

**Author Contributions:** Conceptualization, H.I.B.-R., E.C.-B. and M.d.l.L.X.N.-R.; Data curation, H.I.B.-R. and M.d.l.L.X.N.-R.; Formal analysis, H.I.B.-R. and E.C.-B.; Funding acquisition, M.d.l.L.X.N.-R.; Investigation, S.L.G.-D., F.P.G.-V., D.Á.-B. and M.d.l.L.X.N.-R.; Methodology, H.I.B.-R., E.C.-B., F.P.G.-V., D.Á.-B. and M.d.l.L.X.N.-R.; Project administration, M.d.l.L.X.N.-R.; Resources, H.I.B.-R., Eloy Conde-Barajas, F.P.G.-V., D.Á.-B. and M.d.l.L.X.N.-R.; Software, H.I.B.-R.; Supervision, H.I.B.-R., Eloy Conde-Barajas, F.P.G.-V., D.Á.-B. and M.d.l.L.X.N.-R.; Validation, H.I.B.-R., E.C.-B. and M.d.l.L.X.N.-R.; Visualization, H.I.B.-R. and E.C.-B.; Writing—original draft, H.I.B.-R. and M.d.l.L.X.N.-R.; Writing—review & editing, M.d.l.L.X.N.-R. All authors have read and agreed to the published version of the manuscript.

**Funding:** This research was funded by Tecnológico Nacional de México/IT de Celaya, Gto., Mexico. Tecnológico Nacional de Mexico/IT de Celaya Project, grant number 5721.16.-P. The APCs were covered by Project grant number 5721.16-P.

**Institutional Review Board Statement:** Not applicable.

**Informed Consent Statement:** Not applicable.

**Acknowledgments:** To Municipal Drinking Water and Sewerage Board of the Municipality of Celaya, Guanajuato., Mexico (JUMAPA). Special acknowledgements to Patricia Adriana Estrada Orozco, manager of the sanitation area of JUMAPA, and to José Luis Álvarez Trejo (Died in July 2022) manager of administrative and operation of the wastewater treatment plant (WWTP) of Celaya, Guanajuato. Mexico, for the administrative-technical management facilities in obtaining the urban biosolids used for the present investigation.

**Conflicts of Interest:** The authors declare no conflict of interest.

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
