# Peer review of "Compost Quality Indexes (CQIs) of Biosolids Using Physicochemical, Biological and Ecophysiological Indicators: C and N Mineralization Dynamics"

_agronomy, doi:10.3390/agronomy12102290_

Round 1

Reviewer 1 Report

Dear Authors,

1) In the statistical analysis you used the Tukey test which indicates a parametric ANOVA. Provide information that the normality of the population distribution and the homogeneity of variance in the samples were tested (specify what tests were and what their result was, you will prove that you had the possibility of using parametric ANOVA)

2) Figure 1 - I understand the authors' intentions regarding the graphical presentation of the results. However, in my opinion, the data in a-d charts should not be combined (this indicates data predictions). I suggest that you leave the scatter of points, possibly add a trend line (at the Authors' decision)

3) Figure 1 - give information what the error bars mean in the chart f (5%, Sd, mean, etc.)

4) Notes 2 and 3 above also apply to the remaining drawings

5) Figure 9-12; enlarge the font in the drawing and in the axes.

6) In the applications, indicate the applicability of the obtained results.

7) Consider not including process modeling in the introduction / discussion (e.g. DOI 10.1007 / 978-3-319-72371-6_15)

Author Response

Manuscript ID: agronomy-1928445

Article Type: Research Paper

Title: Compost Quality Indexes (CQIs) of Biosolids Using Physicochemical, Biological And Ecophysiological Indicators: C and N Mineralization Dynamics

September 19, 2022

Reviewer 1

Thank you very much for reviewing the manuscript. The comments you made were valuable and very helpful in improving the research work done. We have carefully read all the comments and have done our best to correct the manuscript according to the suggestions and observations. Finally, we hope to meet the acceptance requirements for publication.

We have included the original comments and responses to them. They have been written in red and are preceded by the word "Response". Also, the corresponding changes in the text of the manuscript have been highlighted using the change control of the Microsoft Word software, as suggested to us, so that you can locate the changes made.

Thank you again for your time and consideration of the revised manuscript.

Sincerely,

The authors.

Response to the reviewer comments

Reviewer #1

Point 1. In the statistical analysis you used the Tukey test which indicates a parametric ANOVA. Provide information that the normality of the population distribution and the homogeneity of variance in the samples were tested (specify what tests were and what their result was, you will prove that you had the possibility of using parametric ANOVA).

Response 1. Taking into consideration the reviewer's recommendation, the normality test used for the data was added. In addition, the wording of the paragraph was improved by adding information regarding the nonparametric analysis of variance used for the quality results obtained through the CQIs developed L.321-323 and L.339-342 respectively.

Point 2. Figure 1 - I understand the authors' intentions regarding the graphical presentation of the results. However, in my opinion, the data in a-d charts should not be combined (this indicates data predictions). I suggest that you leave the scatter of points, possibly add a trend line (at the Authors' decision).

Response 2. We appreciate the comments, however, it should be clarified that the graphs do not represent the modelling or prediction of the indicators analyzed, they are simply a line graph. This type of graph was selected in order to allow the reader to observe the trends in the behavior of the indicators analyzed during the mineralization dynamics of the composting process. As for the reviewer's suggestion to leave the scatter plot, with this option no trend of the indicators would be observed.

Point 3. Figure 1 - give information what the error bars mean in the chart f (5%, Sd, mean, etc.).

Response 3. The review of the figure caption of figure 1 was made, noting that the information requested by the reviewer is in it L701, adding the information requested in the continuations of figure 1 L710, L715.

Point 4. Notes 2 and 3 above also apply to the remaining drawings.

Response 4. All figures were revised as requested by the reviewer and the information requested in Figure 2 L769, Figure 4 L933-934 and Figure 5 L1011 was added.

Point 5. Figure 9-12; enlarge the font in the drawing and in the axes.

Response 5. The images were reviewed, and enlargements were made to the size of the axis titles and scales of figures 9 to 12.

Point 6. In the applications, indicate the applicability of the obtained results.

Response 6. At the request of the reviewer, a subsection focusing on the applications, challenges and prospects of the study has been added L1115-1126.

Point 7. Consider not including process modeling in the introduction / discussion (e.g. DOI 10.1007 / 978-3-319-72371-6_15).

Response 7. We are grateful for the comments, however, it should be clarified that no modelling was performed. The studies by Saldarriaga, Peña and Meena were included in the discussion and introduction for comparison with the present study, since they partially use the same methodology.

Reviewer 2 Report

The manuscript " Compost Quality Indexes (CQIs) of Biosolids Using Physicochemical, Biological And Ecophysiological Indicators: C and N Mineralization Dynamics” analyzed the composting systems using biosolids as the main substrate for the establishment of physicochemical, biological and ecophysiological indicators related to compost quality. The manuscript is original and very interesting, however there are some concerns and points which must be improved. Authors must work on the following points:

* The abstract should be revised. The authors should briefly discuss the purpose of the research and mention their findings adopted in this study. More quantitative data needs to be provided in the abstract.

*Provide significant words which are more relevant to the work in logical sequence as ‘keywords’.

* What is the current level of understanding in relation to the use of biosolids as soil conditioner? What are the knowledge gaps?. These should be included in the introduction section. The introduction is insufficient to provide the state of the art in the topic. Hypothesis should be given. How this work is different from the available data?

*The originality and novelty of the paper need to be further clarified. What progress against the most recent state-of-the-art similar studies was made in this study?

* why authors used bovine manure and rice husks, any background information?

* The discussion and interpretation of results does not clearly explain its impact on the literature and the field. Authors are suggested to add discussion by explaining trends in the obtained results along with the possible mechanisms behind the trends.

* It is strongly recommended to add a subsection, 'practical implications of this study,' outlining the challenges in the current research, future work, and recommendations, before the conclusion.

* Pls. conclude with more focus on the major outcomes of the paper.  

* Check and correct grammatical and space errors throughout the article.

Author Response

Manuscript ID: agronomy-1928445

Article Type: Research Paper

Title: Compost Quality Indexes (CQIs) of Biosolids Using Physicochemical, Biological And Ecophysiological Indicators: C and N Mineralization Dynamics

September 19, 2022

Reviewer 2

Thank you very much for reviewing the manuscript. The comments you made were valuable and very helpful in improving the research work done. We have carefully read all the comments and have done our best to correct the manuscript according to the suggestions and observations. Finally, we hope to meet the acceptance requirements for publication.

We have included the original comments and responses to them. They have been written in red and are preceded by the word "Response". Also, the corresponding changes in the text of the manuscript have been highlighted using the change control of the Microsoft Word software, as suggested to us, so that you can locate the changes made.

Thank you again for your time and consideration of the revised manuscript.

Sincerely,

The authors.

Response to the reviewer comments

Reviewer #2

Point 1. The abstract should be revised. The authors should briefly discuss the purpose of the research and mention their findings adopted in this study. More quantitative data needs to be provided in the abstract.

Response 1. In accordance with the recommendations made by the reviewer, the summary was modified, the treatments implemented were established, as well as the time of the composting process. In addition, the average quality results obtained by the different CQI developed L.18-25 and L28-32 were also included.

Point 2. Provide significant words which are more relevant to the work in logical sequence as keywords.

Response 2. The key words were rearranged, words such as quality indicators, multivariate analysis were eliminated and those related to enzymatic activity and C and N transformation were included, which were considered more appropriate and related to the study L33-34.

Point 3. What is the current level of understanding in relation to the use of biosolids as soil conditioner? What are the knowledge gaps? These should be included in the introduction section. The introduction is insufficient to provide the state of the art in the topic. Hypothesis should be given. How this work is different from the available data?

Response 3. Modifications were made according to the reviewer's request, increasing the information regarding the effects of soil improvement by the addition of biosolids L.52-57. At the same time, information was added regarding the differences of the present study in comparison with previous studies, related to the lack of indicators directly focused on the quality of biosolids compost L86-88. Finally, the research hypothesis was completed L95-96.

Point 4. The originality and novelty of the paper need to be further clarified. What progress against the most recent state-of-the-art similar studies was made in this study?

Response 4. Taking into account the reviewer's recommendations, relevant information was added regarding the differences of the present study with respect to previous studies related to biosolids composting L86-92.

Point 5. why authors used bovine manure and rice husks, any background information?

Response 5. The use of cattle manure has been implemented in composting processes, mainly due to its contribution of C and N compounds, as well as the microbial load present, improving nutrient cycling when added to soils. [Luo, G., Li, L., Friman, V.-P., Guo, J., Guo, S., Shen, Q., & Ling, N. (2018). Organic amendments increase crop yields by improving microbe-mediated soil functioning of agroecosystems: A meta-analysis. Soil Biology and Biochemistry, 124, 105-115. https://doi.org/10.1016/j.soilbio.2018.06.002]. In reference to the use of rice husks in composting systems, its use as a bulking agent has been reported, which improves the conditions of bulk density, aeration and water retention. This would improve the physicochemical conditions for the development of microbial communities that allow the efficient degradation of organic matter in the composting system, thus improving its quality.  [Luo, G., Li, L., Friman, V.-P., Guo, J., Guo, S., Shen, Q., & Ling, N. (2018). Organic amendments increase crop yields by improving microbe-mediated soil functioning of agroecosystems: A meta-analysis. Soil Biology and Biochemistry, 124, 105-115. https://doi.org/10.1016/j.soilbio.2018.06.002]. [Sciubba, L., Cavani, L., Marzadori, C., & Ciavatta, C. (2013). Effect of biosolids from municipal sewage sludge composted with rice husk on soil functionality. Biology and Fertility of Soils, 49(5), 597-608. https://doi.org/10.1007/s00374-012-0748-4]

Point 6. The discussion and interpretation of results does not clearly explain its impact on the literature and the field. Authors are suggested to add discussion by explaining trends in the obtained results along with the possible mechanisms behind the trends.

Response 6. Taking into consideration the reviewer's suggestions, modifications were made to the text L691-695, L1066-1071, L1075-1076, L1080-1081 and L1083-1084.

Point 7. It is strongly recommended to add a subsection, 'practical implications of this study,' outlining the challenges in the current research, future work, and recommendations, before the conclusion.

Response 7. At the request of the reviewer, a subsection focusing on the applications, challenges and prospects of the study has been added L1115-1126.

Point 8. Pls. conclude with more focus on the major outcomes of the paper.

Response 8. Taking into consideration the reviewer's comments, modifications were made to the conclusion section, eliminating redundant sections and adding information regarding the applicability of the study L1138-1141.

Point 9. Check and correct grammatical and space errors throughout the article.

Response 9. In reference to the reviewer's comments, a thorough review of the paper in general was again performed.

Round 2

Reviewer 2 Report

The authors addressed all the comments, therefore the manuscript may be accepted in the present form.